# Somatic aging pathways regulate reproductive plasticity in *Caenorhabditis elegans*

**Maria C Ow, Alexandra M Nichitean, Sarah E Hall\***

Department of Biology, Syracuse University, Syracuse, United States

**Abstract** In animals, early-life stress can result in programmed changes in gene expression that can affect their adult phenotype. In *C. elegans* nematodes, starvation during the first larval stage promotes entry into a stress-resistant dauer stage until environmental conditions improve. Adults that have experienced dauer (postdauers) retain a memory of early-life starvation that results in gene expression changes and reduced fecundity. Here, we show that the endocrine pathways attributed to the regulation of somatic aging in *C. elegans* adults lacking a functional germline also regulate the reproductive phenotypes of postdauer adults that experienced early-life starvation. We demonstrate that postdauer adults reallocate fat to benefit progeny at the expense of the parental somatic fat reservoir and exhibit increased longevity compared to controls. Our results also show that the modification of somatic fat stores due to parental starvation memory is inherited in the $F_1$ generation and may be the result of crosstalk between somatic and reproductive tissues mediated by the germline nuclear RNAi pathway.

**\*For correspondence:**
shall@syr.edu

**Competing interests:** The authors declare that no competing interests exist.

## Introduction

Evidence indicating that experiences during early development affect behavior and physiology in a stress-specific manner later in life is abundant throughout the animal kingdom (*Telang and Wells, 2004*; *Weaver et al., 2004*; *Binder et al., 2008*; *Pellegroms et al., 2009*; *van Abeelen et al., 2012*; *Zhao and Zhu, 2014*; *Canario et al., 2017*; *Dantzer et al., 2019*; *Vitikainen et al., 2019*). Epidemiological studies and experiments using mammalian animal models have supported the 'thrifty' phenotype hypothesis which proposes that fetal or postnatal malnutrition results in increased risk for metabolic disorders in the offspring (*Neel, 1962*; *Hales and Barker, 1992*; *Vaag et al., 2012*; *Smith and Ryckman, 2015*). For instance, individuals exposed to the WWII Dutch Hunger Winter during gestation had lower glucose tolerance and increased risk of obesity, diabetes, and cardiovascular diseases in adulthood compared to siblings born before the famine. In addition, the increased propensity to develop metabolic disorders was inherited for two generations (*Painter et al., 2008*; *Lumey et al., 2011*; *Veenendaal et al., 2013*). Thus, stress, such as malnutrition, early in life and the ensuing metabolic and physiological adaptation highlight the effectiveness by which the environment reconfigures animal life history.

The nematode *C. elegans* makes a critical decision regarding its developmental trajectory based on the environmental conditions experienced during its early larval stages (L1-L2). If conditions are poor (e.g. low food availability, crowding, or high temperatures), decreased insulin and TGF-β signaling promote entry into an alternative, stress-resistant, non-aging, diapause stage named dauer. Once conditions improve, dauer larvae resume development as postdauer L4 larvae and continue through reproductive adulthood as postdauer adults (*Cassada and Russell, 1975*). Alternatively, if conditions are favorable, L1 larvae proceed through additional larval stages (L2-L4) until reaching reproductive adulthood (control adults) (*Sulston and Horvitz, 1977*). Although postdauer adults are morphologically similar to control adults, we previously showed that postdauer adults retained a

cellular memory of their early-life experience that resulted in genome-wide changes in their chromatin, transcriptome, and life history traits (*Hall et al., 2010*; *Hall et al., 2013*; *Ow et al., 2018*). Remarkably, postdauer adults also encoded the nature of their early environmental stress and gauged their adult reproductive phenotypes and genome-wide gene expression based on this memory (*Ow et al., 2018*). Postdauer adults that experienced crowding or high pheromone conditions exhibited increased fecundity and upregulated expression of genes involved in reproduction relative to control adults that never experienced crowding. In contrast, postdauer adults that experienced starvation ($PD_{Stv}$) exhibited decreased fecundity and an enrichment in somatic gene expression compared to control adults that never experienced starvation (CON) (*Ow et al., 2018*). Moreover, the changes in fecundity and somatic gene expression in $PD_{Stv}$ adults required a functional germ line (*Ow et al., 2018*).

The crosstalk pre-requisite between somatic and reproductive tissues for postdauer reproductive phenotypes is also a key regulatory feature governing adult lifespan and stress response (*Kenyon, 2010a*; *Kenyon, 2010b*). In *C. elegans*, endocrine signaling has emerged as one of the principal pathways extending the lifespan of animals lacking a germ line due to either ablation of germ line precursor cells or mutations in the *glp-1*/Notch receptor gene. The two main effectors of endocrine signaling, the FOXO transcription factor DAF-16 and the nuclear hormone receptor (NHR) DAF-12, are required for the increased lifespan of germ line-less animals and are regulated by the physiological state of the animal (*Hsin and Kenyon, 1999*; *Kenyon, 2010a*; *Kenyon, 2010b*; *Murphy and Hu, 2013*). When an animal experiences reproductive stress, such as sterility, DAF-16 is dephosphorylated and translocated to the nucleus where it can modify target gene expression to promote the extended lifespan of germ line-less animals (*Kenyon, 2010a*; *Kenyon, 2010b*; *Murphy and Hu, 2013*). DAF-12, a homolog of the mammalian vitamin D receptor, binds to bile acid-like steroid ligands (dafachronic acids or DA) to boost the expression of genes involved in reproduction and growth under favorable conditions (*Antebi, 2014*). In *glp-1* mutants, the $\Delta^7$ form of DA ($\Delta^7$-DA) is increased fourfold compared to wild type and promotes DAF-16 nuclear localization (*Shen et al., 2012*). One of the consequences of the DAF-16 and DAF-12-dependent endocrine signaling in *glp-1* mutants is a significant increase in stored intestinal fat, which allows for somatic maintenance and prolonged lifespan in the absence of germline development (*Wang et al., 2008*).

In this study, we show that steroid hormone signaling, reproductive longevity signaling, and nuclear hormone receptors contribute to the decreased fecundity of postdauer animals that experienced early-life starvation by modifying fatty acid metabolism. The reproductive plasticity of $PD_{Stv}$ adults is a result of crosstalk between somatic and reproductive tissues, the effect of which is an increase in lipid metabolic pathway function in an animal that has experienced dauer, resulting in decreased lipid storage in the adult and reallocation of fat into embryos. Thus, the pathways that bestow increased lipid storage and extended longevity in a germ line-less animal function to promote reproduction in a postdauer animal that experienced early-life starvation. We also show that the $F_1$ generation inherits the parental memory for altered fat metabolism manifested as increased intestinal fat storage, which is dependent on HRDE-1 and PRG-1, two germline-specific RNAi Argonautes. Given the role of these Argonautes in RNAi-mediated transgenerational inheritance, our results suggest that RNAi pathways may transmit an ancestral starvation memory through the modulation of fat metabolism to ensuing generations to provide the necessary hardiness to survive future famine.

## Results

### Dafachronic acid-dependent DAF-12 signaling may be required for decreased fecundity after starvation-induced dauer formation

Given that endocrine signaling across tissues is a prominent feature of reproductive longevity, we examined whether wild-type $PD_{Stv}$ adults shared any gene expression signatures with animals lacking a functional germ line. In *glp-1* mutants, increased longevity is dependent on TOR (target of rapamycin) signaling, DAF-16/FOXO gene regulation, steroid hormone signaling, and fatty acid metabolism regulation (*Lapierre and Hansen, 2012*). With the exception of TOR signaling, we found significant gene expression changes in $PD_{Stv}$ adults of key genes in each of these regulatory pathways (*Supplementary file 1*).

In the steroid signaling pathway, dafachronic acid (DA) biosynthesis requires the cytochrome P450 DAF-9, the Rieske-like oxygenase DAF-36, the short-chain dehydrogenase DHS-16, and the hydroxysteroid dehydrogenase HSD-1 (*Figure 1A*; *Gerisch et al., 2001*; *Jia et al., 2002*; *Motola et al., 2006*; *Rottiers et al., 2006*; *Patel et al., 2008*; *Wollam et al., 2012*; *Mahanti et al., 2014*). In animals lacking a functional germ line, the levels of *daf-36* mRNA and $\Delta^7$-DA are significantly increased compared to wild type (*Shen et al., 2012*). We found that in wild-type PD$_{Stv}$ adults with a germ line, *daf-36* mRNA increased threefold ($p$ = 3.25e-04; FDR = 0.01) compared to control adults (*Supplementary file 1*; *Ow et al., 2018*). To investigate whether DA signaling plays a role in mediating reproductive plasticity as a result of early-life experience, we asked whether mutations in DA biosynthesis genes would affect the reduced brood size observed in PD$_{Stv}$ adults. We found that *daf-9(rh50)*, *daf-36(k114)*, and *dhs-16(tm1890)* mutants did not exhibit a significant decrease in brood size in PD$_{Stv}$ adults compared to controls (CON), while *hsd-1(mg433)* brood sizes were similar to wild type. Interestingly, *daf-9* and *daf-36* mutants exhibited a significant increase in postdauer brood size compared to controls, opposite of what we observed in wild type (*Figure 1B*).

We next asked whether the steroid signaling that contributed to the reduced fecundity in postdauers acted through two related NHRs, DAF-12 and NHR-8. Since null mutants of *daf-12* are dauer defective, we used two *daf-12* mutant alleles, *rh284* (Class 5) and *rh285* (Class 4), with lesions in the ligand binding domain that affect steroid signaling activity but otherwise have a relatively normal dauer phenotype (*Antebi et al., 2000*). We found that *daf-12(rh284)* and *daf-12(rh285)* exhibited a significant increase in PD$_{Stv}$ brood size compared to controls, similar to that observed for the *daf-9* and *daf-36* DA biosynthesis mutants (*Figure 1B*). The closest gene relative to *daf-12*, *nhr-8*, is also upregulated 2.5-fold in PD$_{Stv}$ adults (*Lindblom et al., 2001*; *Magner et al., 2013*; *Ow et al., 2018*); however, *nhr-8(ok186)* adults exhibited a significant brood size reduction in PD$_{Stv}$ animals compared to controls similar to wild type (*Figure 1B*). One possible explanation of this observation is that passage through dauer may partially rescue the reproductive phenotypes of the *daf-12*, *daf-36*, and *daf-9* mutants, as has been previously observed for hypodermal and vulval precursor cell fates in postdauer heterochronic mutants (*Liu and Ambros, 1991*; *Euling and Ambros, 1996*), or that DAF-12 activity in the absence of DA is sufficient for reproduction in PD$_{Stv}$ animals. Another explanation is that DA-dependent DAF-12 activity is required for the early-life starvation memory that programs a decrease in PD$_{Stv}$ fecundity, and its loss results in a reproductive phenotype similar to what we have observed previously in pheromone-induced postdauers (*Ow et al., 2018*). Although we cannot distinguish between these possibilities with this data, we favor the latter explanation given the abrogation of the decreased fecundity phenotype in *dhs-16* mutants, which lack significant reproductive defects as determined by the similarity of the brood size of *dhs-16* control adults to wild type (*Figure 1B*).

## The TCER-1 reproductive longevity pathway mediates reproductive plasticity

DAF-16 and PQM-1 act in a mutually antagonistic manner to promote the expression of a group of stress response genes (Class I) or genes associated with growth and reproduction (Class II), respectively (*Figure 2A*; *Tepper et al., 2013*). We found that the set of genes with significant changes in mRNA levels between PD$_{Stv}$ and controls was enriched for Class I and II targets (*Figure 2—figure supplement 1A*; *Figure 2—figure supplement 1—source data 1*). In addition, we found two genes that regulate DAF-16 cellular localization, *pqm-1* and *daf-18*, were significantly up and downregulated, respectively, in PD$_{Stv}$ adults compared to controls (*Supplementary file 1*; *Ow et al., 2018*). DAF-18 is the functional ortholog of the human PTEN tumor suppressor gene that promotes DAF-16 nuclear localization (*Ogg and Ruvkun, 1998*; *Gil et al., 1999*; *Mihaylova et al., 1999*; *Solari et al., 2005*). These observations prompted us to ask whether PQM-1 and DAF-18 contribute to the reduced fertility in PD$_{Stv}$ adults by altering the regulation of genes involved in reproduction and sequestering DAF-16 in the cytoplasm. However, the brood size differences between control and PD$_{Stv}$ adults in *pqm-1(ok485)* and *daf-18(e1375)* mutants were similar to wild type, indicating that gene regulation by PQM-1 is unlikely to contribute to the PD$_{Stv}$ reduced fecundity (*Figure 2B*). Next, because *daf-16* null mutants are dauer defective, we crossed a *daf-16(mu86)* null allele with a strain carrying a rescue transgene (*daf-16a^{AM}::gfp*) that constitutively localizes DAF-16 to the nucleus (*Lin et al., 2001*). Similar to what was observed for wild type, the *daf-16(mu86); daf-16a^{AM}::gfp* transgenic strain displayed a reduced brood size in PD$_{Stv}$ compared to controls (*Figure 2B*). Since

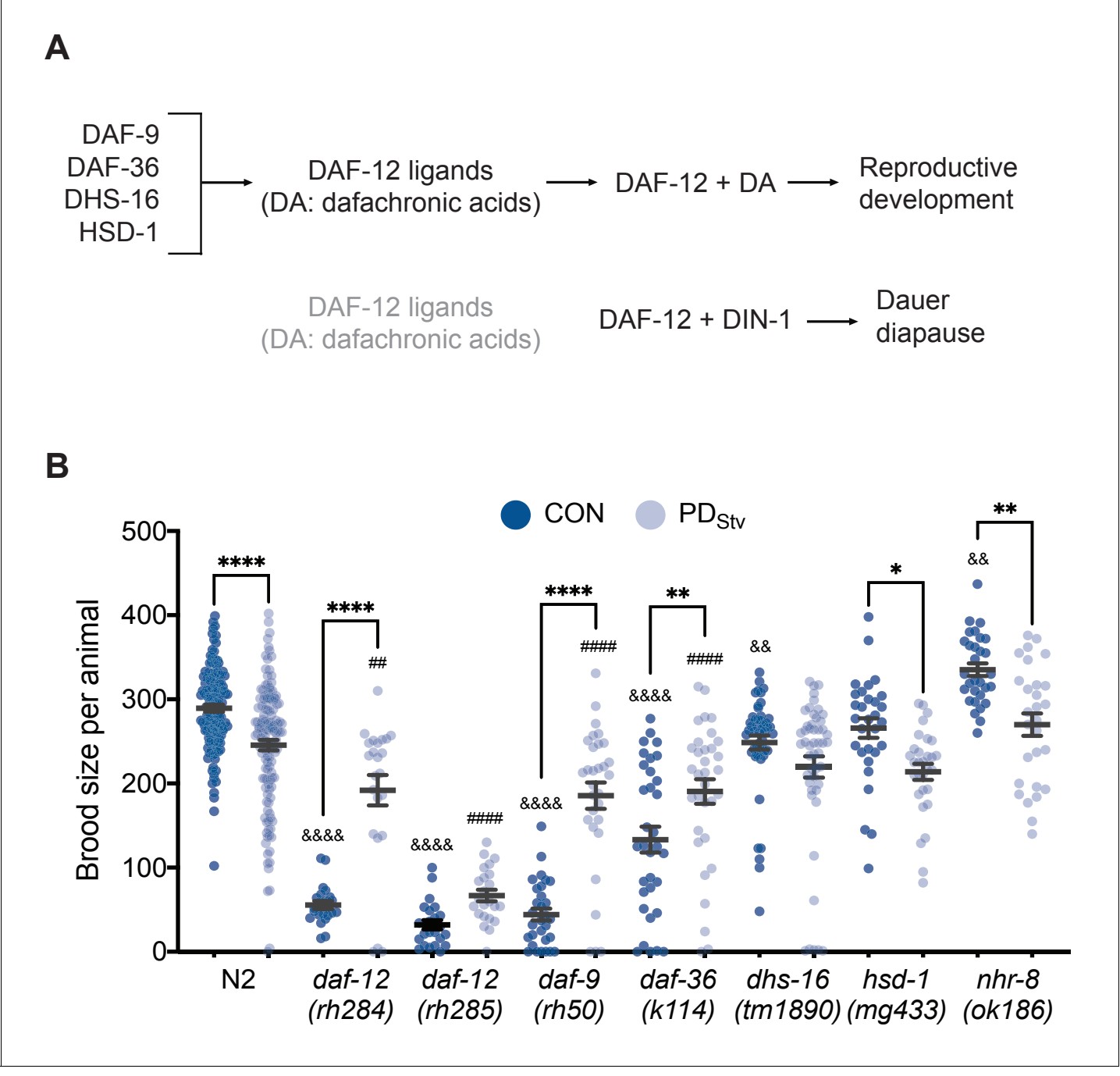

**Figure 1.** Adult reproductive plasticity is dependent on DAF-12 steroid signaling. (**A**) Model of DAF-12 regulation of development. See text for details. (**B**) Brood size of CON and PD$_{Stv}$ in wild-type N2 and mutant strains. * $p < 0.05$, ** $p < 0.01$, and **** $p < 0.0001$ compare CON and PD$_{Stv}$ within a strain; $^{\&\&}p < 0.01$ and $^{\&\&\&\&}p < 0.0001$ compare N2 CON to mutant CON; $^{\#\#}p < 0.01$ and $^{\#\#\#\#}p < 0.0001$ compare N2 PD$_{Stv}$ to mutant PD$_{Stv}$; one-way ANOVA with Sidak's multiple comparison test. Error bars represent S.E.M. Additional data are provided in *Figure 1—source data 1*.

The online version of this article includes the following source data and figure supplement(s) for figure 1:

**Source data 1.** Dafachronic acid-dependent DAF-12 signaling is required for decreased fecundity after starvation-induced dauer formation.

**Figure supplement 1.** Dafachronic acid affects adult reproductive plasticity.

**Figure supplement 1—source data 1.** Brood size comparison between PDStv grown in the presence of delta7-dafachronic acid (PDStv + DA) and PDStv grown in the absence of (delta7-DA (PDStv - DA).

**Figure supplement 1—source data 2.** Brood size comparison of PDStv relative to CON.

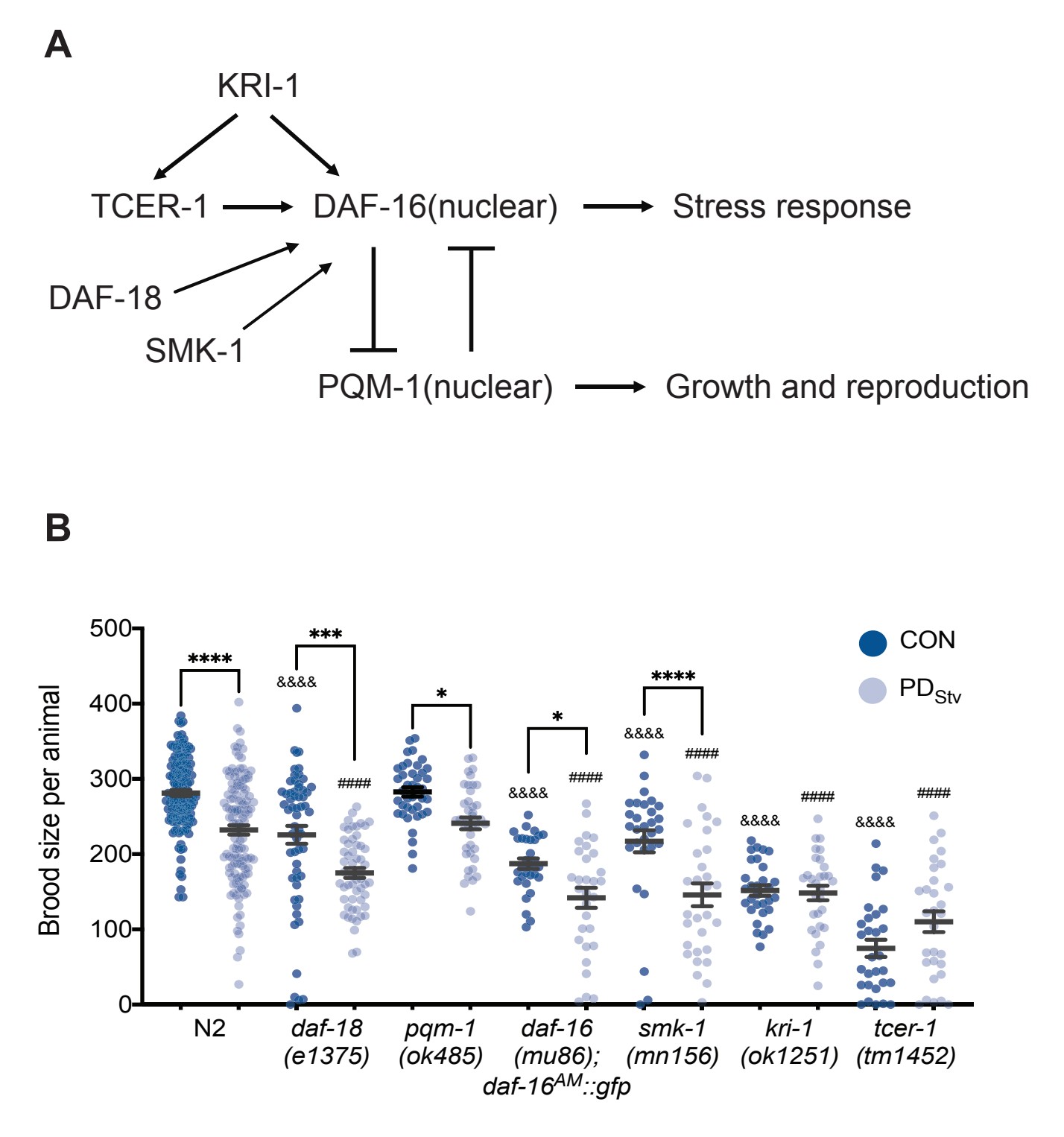

**Figure 2.** TCER-1 and KRI-1 regulate the decreased fecundity phenotype in PD_Stv adults. (**A**) Model of the regulation of DAF-16 nuclear localization. See text for details. (**B**) Brood size of CON and PD_Stv in wild-type N2, *daf-18(e1375)*, *pqm-1(ok485)*, *daf-16(mu86); daf-16^{AM}::gfp*, *smk-1(mn156)*, *kri-1 (1251)*, and *tcer-1(tm1452)*. * $p < 0.05$, *** $p < 0.001$, and **** $p < 0.0001$ compare CON and PD_Stv within a genotype; &&&& $p < 0.0001$ compares N2 CON to mutant CON; #### $p < 0.0001$ compares N2 PD_Stv to mutant PD_Stv; one-way ANOVA with Sidak's multiple comparison test. Error bars represent S.E.M. Additional data are provided in **Figure 2—source data 1**.

The online version of this article includes the following source data and figure supplement(s) for figure 2:

*Figure 2 continued on next page*

Figure 2 continued

**Source data 1.** TCER-1 and KRI-1 regulate the decreased fecundity phenotype in PDStv adults.
**Figure supplement 1.** Differentially expressed genes in wild-type N2 PD$_{Stv}$ are enriched for DAF-16 targets.
**Figure supplement 1—source data 1.** Comparison of DAF-16 and PQM-1 targets (*Tepper et al., 2013*) with differentially expressed genes in N2 PDStv to N2 CON (*Ow et al., 2018*).
**Figure supplement 1—source data 2.** Comparison of DAF-16 and TCER-1 targets with functions in lipid metabolism from germ line-less glp-1(e2141ts) animals (*Amrit et al., 2016*) with differentially expressed genes in N2 PDStv to N2 CON (*Ow et al., 2018*).
**Figure supplement 2.** Localization of DAF-16::GFP in CON and PD$_{Stv}$ one-day-old adults in (A) CF1139 (*daf-16(mu86) I; muIs61 [(pKL78) daf-16::gfp + rol-6(su1006)]*) and (B) TJ356 (*zIs356 [daf-16p::daf-16a/b::gfp + rol-6(su1006)]*).

*daf-18(e1375)* is a hypomorph, we next tested the possibility that DAF-16 nuclear localization may play a role in regulating postdauer reproduction. First, we tested SMK-1/PPP4R3A, which promotes the nuclear localization of DAF-16 when animals are exposed to pathogenic bacteria, ultraviolet irradiation, or oxidative stress (*Wolff et al., 2006*), and found that *smk-1(mn156)* mutants continued to exhibit a decreased PD$_{Stv}$ fertility compared to controls (*Figure 2B*). In addition, we examined the cellular localization of *daf-16*p::*daf-16a/b::gfp* transgene in two independent strains (CF1139 and TJ356) and observed diffuse cytoplasmic signal in intestinal cells of both control and PD$_{Stv}$ adults (*Figure 2—figure supplement 2*). Together, these results do not support a role for DAF-16 per se in the diminished fertility phenotype in postdauers.

In the reproductive longevity pathway, two genes, *kri-1* (ortholog of the human intestinal ankyrin-repeat protein KRIT1/CCM1) and *tcer-1* (homolog of the human transcription elongation factor TCERG1), are required for DAF-16 nuclear localization and increased longevity in germ line-less animals (*Berman and Kenyon, 2006*; *Ghazi et al., 2009*). TCER-1 regulates target genes in both a DAF-16-dependent and independent manner (*Amrit et al., 2016*). Since we determined that DAF-16 itself does not contribute to the PD$_{Stv}$ reproductive phenotype, we tested whether TCER-1 and KRI-1 regulate PD$_{Stv}$ reproduction independent of DAF-16. Interestingly, we found that the decreased fecundity in PD$_{Stv}$ was abrogated in *kri-1(ok1251)* and *tcer-1(tm1452)* strains (*Figure 2B*), indicating that KRI-1 and TCER-1 are required for the reproductive plasticity in PD$_{Stv}$ animals in a DAF-16-independent manner.

## Increased fatty acid metabolism promotes PD$_{Stv}$ fertility

In animals lacking a germ line, DAF-16 and TCER-1 are required to bolster the expression of lipid biosynthesis, storage, and hydrolysis genes to promote adult longevity (*Amrit et al., 2016*). Similar to germ line-less *glp-1* mutants, PD$_{Stv}$ adults exhibited a significantly altered expression of ~26% (33 of 127 genes) of all the fatty acid metabolic genes, including ~46% (18 of 39 genes) also targeted by DAF-16 and TCER-1 (*Figure 2—figure supplement 1B*; *Figure 2—figure supplement 1—source data 2*). To investigate whether reduced fecundity of PD$_{Stv}$ is modulated by upregulating fatty acid metabolism, we asked if mutations in known regulators of lipid metabolism would exhibit changes in brood size in PD$_{Stv}$ adults when compared to controls. One of the genes jointly upregulated by DAF-16 and TCER-1 is *nhr-49*, a nuclear hormone receptor homologous to the mammalian HNF4α lipid sensing nuclear receptor involved in the regulation of fatty acid metabolism and the oxidative stress response (*Ratnappan et al., 2014*; *Amrit et al., 2016*; *Moreno-Arriola et al., 2016*; *Goh et al., 2018*; *Hu et al., 2018*). Additional nuclear hormone receptors, NHR-80, NHR-13, and NHR-66, and the Mediator complex subunit MDT-15, partner with NHR-49 and co-regulate the expression of genes involved in various aspects of lipid metabolism such as fatty acid β-oxidation, transport, remodeling, and desaturation (*Gilst et al., 2005*; *Van Gilst et al., 2005*; *Taubert et al., 2006*; *Nomura et al., 2010*; *Pathare et al., 2012*; *Ratnappan et al., 2014*; *Folick et al., 2015*; *Amrit et al., 2016*). In addition, SBP-1 (homolog of mammalian SREBP) and NHR-49 are co-regulated by MDT-15 as part of a transcriptional network coordinating the expression of delta-9 (Δ9) fatty acid desaturase genes (*Figure 3—figure supplement 1A*; *Yang et al., 2006*; *Taubert et al., 2006*).

When we examined the control and PD$_{Stv}$ brood sizes of strains carrying mutations in fatty acid metabolism genes, we found that the reduced fecundity of PD$_{Stv}$ characteristic of wild-type animals was also observed in *nhr-80(tm1011)*, *nhr-13(gk796)*, and in the *nhr-49(ok2165)* allele (*Figure 3A*). Postdauers expressing *nhr-49* gain-of-function (*gf*) alleles *et7* or *et13* also exhibited reduced brood size (*Figure 3A*). However, *nhr-66(ok940)*, *mdt-15(tm2182)*, and *sbp-1(ep79)* strains, in addition to

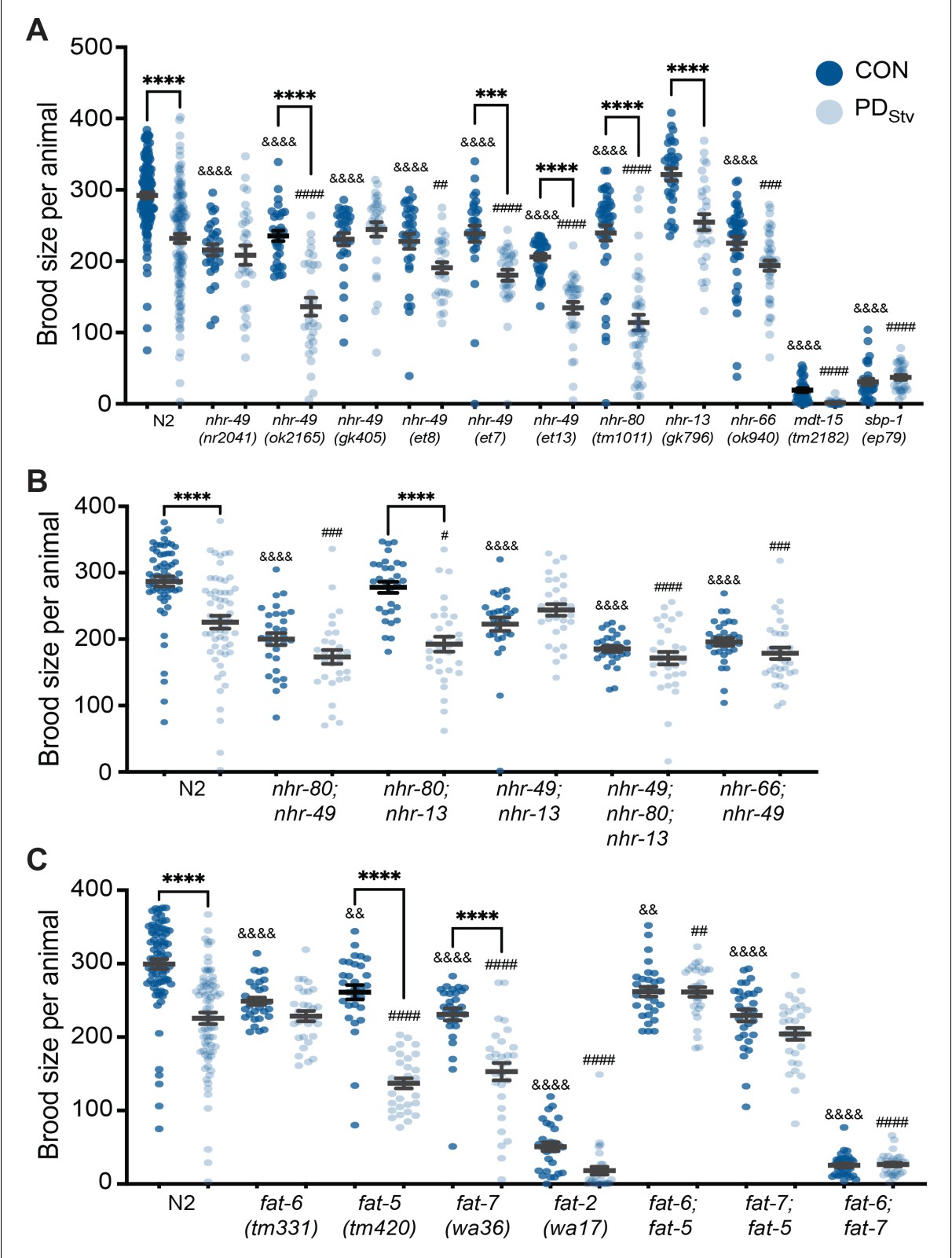

**Figure 3.** Fatty acid metabolism pathways modulate adult reproductive plasticity. (A, B, C) Brood sizes of CON and PD$_{Stv}$ in wild-type N2 and mutant strains. *** $p < 0.001$ and ****$p < 0.0001$ compare CON and PD$_{Stv}$ within a genotype; $^{\&\&}p < 0.01$ and $^{\&\&\&\&}$ $p < 0.0001$ compare N2 CON to mutant CON; $^{\#\#}$ $p < 0.01$, $^{\#\#\#}$ $p < 0.001$, and $^{\#\#\#\#}$ $p < 0.0001$ compare N2 PD$_{Stv}$ to mutant PD$_{Stv}$; one-way ANOVA with Sidak's multiple comparison test. Error bars represent S.E.M. Additional data are provided in *Figure 3—source data 1*, *Figure 3—source data 2*, and *Figure 3—source data 3*.

*Figure 3 continued on next page*

*Figure 3 continued*

The online version of this article includes the following source data and figure supplement(s) for figure 3:

**Source data 1.** Fatty acid metabolism pathways modulate adult reproductive plasticity.
**Source data 2.** Fatty acid metabolism pathways modulate adult reproductive plasticity.
**Source data 3.** Fatty acid metabolism pathways modulate adult reproductive plasticity.
**Figure supplement 1.** Fatty acid metabolism in C. elegans.

---

three *nhr-49* alleles, *nr2041*, *gk405*, and the *et8 gf*, failed to exhibit the decreased fecundity in $PD_{Stv}$ adults compared to CON (*Figure 3A*). The *gf nhr-49* alleles, *et7*, *et8*, and *et13*, harbor missense lesions located at or near the ligand-binding domain (*Svensk et al., 2013*; *Lee et al., 2016*). The nature of *et7* and *et13* could modify NHR-49 activity following the dauer experience, resulting in a significant decrease in postdauer brood size (*Figure 3A*). Because the six *nhr-49* mutant alleles differ in the nature and the location of their lesions, their physiological function could vary and result in disparate reproductive phenotypes.

Amongst the various *nhr-49* alleles that eliminated the $PD_{Stv}$ reproductive phenotype, we chose to use the well-characterized *nr2041* for further experiments owing that it is a complete loss-of-function mutant whose lesion encompasses a deletion in its DNA binding domain as well as over half of its ligand-binding domain (*Gilst et al., 2005*; *Van Gilst et al., 2005*; *Pathare et al., 2012*). Because of the interaction between NHR-49, NHR-80, NHR-13, and NHR-66, we also examined if double and triple mutants of these NHRs would have any reproductive plasticity phenotypes. Strains carrying mutations in *nhr-80*, *nhr-13*, or *nhr-66* in addition to *nhr-49* showed an abrogated phenotype compared to wild type (*Figure 3B*). A triple mutant strain, *nhr-49; nhr-80; nhr-13*, showed a similar abrogated phenotype (*Figure 3B*). In contrast, the *nhr-80; nhr-13* double mutant exhibited a wild-type phenotype, indicating the importance of NHR-49 in regulating $PD_{Stv}$ brood size (*Figure 3B*). Together, these data suggest that SBP-1, MDT-15, NHR-49, and interacting NHR, NHR-66, are important in the postdauer reproduction program, likely by upregulating fat metabolism genes.

NHR-49, MDT-15, and SBP-1 upregulate the expression of genes involved in fatty acid biosynthesis, including the Δ9 desaturases, *fat-5*, *fat-6*, *fat-7*, and the delta-12 (Δ12) desaturase *fat-2* (*Yang et al., 2006*; *Nomura et al., 2010*; *Han et al., 2017*). FAT-5, FAT-6, and FAT-7 convert saturated fatty acids (SFAs) to mono-unsaturated fatty acids (MUFAs), while FAT-2 catalyzes the conversion of MUFAs to poly-unsaturated fatty acids (PUFAs) (*Figure 3—figure supplement 1B*; *Watts and Ristow, 2017*). Our previous mRNA-Seq results showed that the expression of *fat-5*, *fat-6*, *fat-7*, and *fat-2* increased significantly between 3.8- and 26.6-fold in wild-type $PD_{Stv}$ adults compared to controls (*Supplementary file 1*; *Ow et al., 2018*). When we compared the $PD_{Stv}$ brood size to controls for these mutant strains, *fat-6* and *fat-2* exhibited an abrogated phenotype, while *fat-5* and *fat-7* strains retained the decreased brood size phenotype similar to wild type (*Figure 3C*). Furthermore, the double mutant strains with combinations of mutations in *fat-5*, *fat-6*, and *fat-7* genes all exhibited an elimination of the decreased brood size phenotype (*Figure 3C*). These results suggest: (1) a functional redundancy between the Δ9 fatty acid desaturases in modulating lipid homeostasis of $PD_{Stv}$ adults, with FAT-6 playing a more principal role than FAT-5 and FAT-7; and (2) MUFA and PUFA levels may be upregulated to promote the decreased fertility phenotype in $PD_{Stv}$ adults compared to controls.

In *C. elegans*, MUFAs are essential for viability as a *fat-5; fat-6; fat-7* triple mutant is lethal (*Brock et al., 2006*). MUFAs, such as oleic acid (OA), can be remodeled to become PUFAs, phospholipids, and neutral lipids such as triacylglycerols (TAG), which serve as energy storage molecules in the intestine, hypodermis, and germ line (*Figure 3—figure supplement 1B*; *Watts and Ristow, 2017*). In addition to acting as key regulators of fat metabolism, FAT-5, FAT-6, and FAT-7 are also essential in promoting the long lifespan of adult worms lacking a germ line (*Gilst et al., 2005*; *Brock et al., 2006*; *Goudeau et al., 2011*; *Ratnappan et al., 2014*). Given that MUFAs may be required for the reduced fecundity of $PD_{Stv}$ adults, we asked whether the dietary addition of OA to $PD_{Stv}$ animals would further reduce their brood size. To test this, we compared the brood size of $PD_{Stv}$ adults fed *E. coli* OP50 grown with OA and $PD_{Stv}$ adults whose bacterial diet was not preloaded with OA. We tested the N2 wild type, *nhr-49*, and Δ9 desaturase double mutant strains. We found that for wild type and the strains that included a mutation in *nhr-49*, the supplementation of

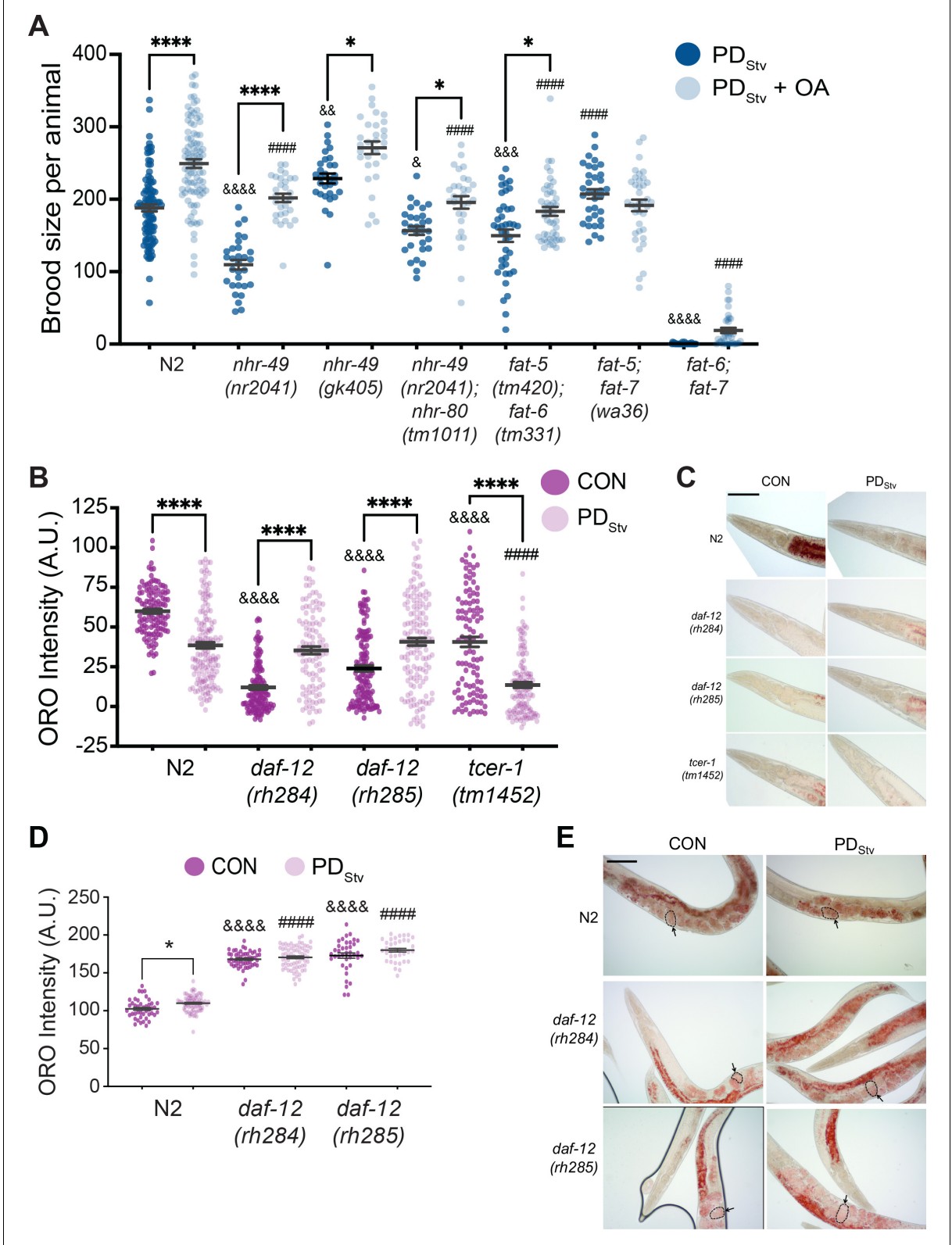

**Figure 4.** Lipid metabolism is affected in wild-type postdauer adults that experienced starvation-induced dauer. (A) Brood size of wild-type N2 PD$_{Stv}$ and mutant PD$_{Stv}$ with or without oleic acid (OA) supplementation. * $p < 0.05$ and **** $p < 0.0001$ compare PD$_{Stv}$ to PD$_{Stv}$ + OA within a genotype; & $p < 0.05$, && $p < 0.01$, &&& $p < 0.001$, and &&&& $p < 0.0001$ compare N2 PD$_{Stv}$ to mutant PD$_{Stv}$; #### $p < 0.0001$ compares of N2 PD$_{Stv}$ + OA to mutant PD$_{Stv}$ + OA; one-way ANOVA with Sidak's multiple comparison test. Additional data are provided in *Figure 4—source data 1*. (B) Oil Red O (ORO) intensity

*Figure 4 continued on next page*

*Figure 4 continued*

in CON and PD$_{Stv}$ one-day-old adults. **** $p < 0.0001$ compares CON and PD$_{Stv}$ of the same genotype; $^{\&\&\&\&}$ $p < 0.0001$ compares N2 CON to mutant CON; $^{\#\#\#\#}$ $p < 0.0001$ compares N2 PD$_{Stv}$ to mutant PD$_{Stv}$; one-way ANOVA with Sidak's multiple comparison test. Error bars represent S.E.M. A.U.: arbitrary units. Additional data are provided in *Figure 4—source data 2*. (C) Representative micrographs of one-day-old adults stained with ORO. Scale bar: 100 µM. (D) ORO intensity of embryos measured in utero in one-day-old adults. * $p < 0.05$ compares CON and PD$_{Stv}$ within a genotype; $^{\&\&\&\&}$ $p < 0.0001$ compares N2 CON to mutant CON; $^{\#\#\#\#}$ $p < 0.0001$ compares PD$_{Stv}$ of N2 to mutant strains; one-way ANOVA with Sidak's multiple comparison test. Error bars represent S.E.M. Additional data is provided in *Figure 4—source data 3*. (E) Representative micrographs of ORO-stained adults. Dotted outlines and arrows are representative of ORO-stained in utero embryos quantified in (D). Scale bar: 100 µM.

The online version of this article includes the following source data and figure supplement(s) for figure 4:

**Source data 1.** Brood size of wild-type N2 PDStv and mutant PDStv with or without oleic acid (OA) supplementation.
**Source data 2.** Oil Red O (ORO) intensity in CON and PDStv one-day-old adults.
**Source data 3.** ORO intensity of embryos measured in utero in one-day-adults.
**Figure supplement 1.** Postdauer one-day-old wild-type N2 adults exhibit fat-stained embryos despite the absence of intestinal fat stores.
**Figure supplement 1—source data 1.** Fatty acid composition of N2 CON and PDStv one-day-old adults.

OA significantly increased PD$_{Stv}$ adult fecundity compared to the control diet (*Figure 4A*). In addition, the *fat-6; fat-5* double mutant strain continued to exhibit a significant increase in brood size when fed food supplemented with OA. However, the Δ9 desaturase double mutant strains carrying a mutation in *fat-7, fat-7; fat-5* and *fat-6; fat-7*, did not exhibit an OA-induced increase in brood size (*Figure 4A*). These results suggest that OA is not required for decreased fecundity but may rather be a limiting factor for reproduction after passage through the starvation-induced dauer stage, whether for nutrition or as a signaling molecule across tissues to regulate physiology (*Schmeisser et al., 2019*; *Starich et al., 2020*). These results also indicate that *fat-7* is required for the OA-induced increase in brood size, which was unexpected given that it acts upstream of OA in fatty acid synthesis and suggests that FAT-7 has additional roles in fatty acid metabolism (*Figure 3—figure supplement 1B*; *Watts, 2009*). Altogether, these results suggest that the upregulation of fatty acid desaturases are critical for the decreased fertility in PD$_{Stv}$ adults by mediating and promoting the synthesis of sufficient levels of lipids needed for reproduction after the animals experienced starvation-induced dauer formation.

## Starvation-induced postdauer adults have reduced fat stores

In long-lived *glp-1* mutants lacking a germ line, fat stores are increased relative to wild type (*Steinbaugh et al., 2015*; *Amrit et al., 2016*). Given that PD$_{Stv}$ adults have an increased expression of lipid metabolism genes similar to *glp-1*, but have limited quantities of OA for reproduction, we questioned what the status of fat stores would be in wild-type PD$_{Stv}$ adults with an intact germ line. Using Oil Red O (ORO) staining, we compared the amounts of neutral triglycerides and lipids (*O'Rourke et al., 2009*) in PD$_{Stv}$ one-day-old adults and developmentally matched controls. Despite having a significant upregulation in fatty acid metabolism genes, we found that PD$_{Stv}$ adults have a significantly reduced amount of stored lipids relative to controls in their intestine (*Figure 4B and C*; *Figure 4—figure supplement 1A and B*). These results are consistent with a model that PD$_{Stv}$ adults have increased expression of lipid metabolism genes for reproduction rather than somatic lipid storage.

Next, we investigated whether the decreased lipid stores in PD$_{Stv}$ adults was dependent upon DAF-12 and TCER-1. DA-dependent DAF-12 activity was shown previously to promote fat utilization for reproduction (*Wang et al., 2015*). In contrast to previous results, we found that control adults of both *daf-12(rh284)* and *daf-12(rh285)* strains displayed low levels of lipid storage, and postdauer adults exhibited a significant increase in lipid storage relative to controls (*Figure 4B and C*; *Figure 4—figure supplement 1B*). Interestingly, the levels of intestinal ORO staining positively correlated with control and PD$_{Stv}$ brood sizes in wild type and the *daf-12* mutants, and the *daf-12* mutant postdauers have statistically similar lipid stores compared to wild-type postdauers, further supporting the conclusion that DA-dependent DAF-12 activity is not required in postdauers to regulate lipid storage and reproduction (*Figures 1B*, *4B and C*; *Figure 4—figure supplement 1B*). In contrast, *tcer-1(tm1452)* lipid staining was diminished in PD$_{Stv}$ compared to controls, similar to wild type (*Figure 4B and C*; *Figure 4—figure supplement 1B*). However, since both *tcer-1(tm1452)* control and PD$_{Stv}$ adults have reduced fat stores compared to their wild-type counterparts (*Figure 4B and*

*C*), and TCER-1 is known to positively regulate NHR-49 and fatty acid metabolism, this result may be due to the inability of these animals to store fat and not because TCER-1 regulates the levels of stored fat in $PD_{Stv}$ adults. These results suggest that fine-tuning the balance of somatic lipid stores between the CON and $PD_{Stv}$ life histories may be correlated with reproductive output.

Given that fatty acid metabolism is important for regulating fecundity in postdauer animals, we profiled fatty acids in wild-type control and $PD_{Stv}$ adults. Quantification of the level of SFAs, MUFAs, and PUFAs revealed that most of these fatty acids, including oleic acid, remained unchanged in control and $PD_{Stv}$ adults. Only two PUFAs, α-linolenic acid (ALA or C18:3n3) and dihomo-γ-linolenic (DGLA or C20:3n6) were significantly downregulated and upregulated, respectively, in $PD_{Stv}$ adults (*Figure 4—figure supplement 1C*). ALA is an omega-3 fatty acid whose level is augmented in *glp-1* animals and is reported to extend adult lifespan in a manner that is dependent on NHR-49 and the SKN-1/Nrf2 transcription factor (*Ratnappan et al., 2014*; *Amrit et al., 2016*; *Qi et al., 2017*). Dietary supplementation of omega-6 fatty acid, DGLA, has been shown to trigger sterility through ferroptosis, an iron-dependent germ line cell death resulting from the production of toxic lipid metabolites (*Deline et al., 2015*; *Perez et al., 2020*). Through the activities of the *fat-2*, *fat-1*, *fat-3*, *elo-1/2* and/or *let-767* genes, oleic acid serves as the substrate for the production of ALA and DGLA (*Figure 3—figure supplement 1B*). Interestingly, with the exception of *fat-2*, we found that the mRNA levels of these genes were significantly increased in postdauers that experienced starvation but not crowded conditions (*Ow et al., 2018*). Thus, the fatty acid profiling suggests that a complex interplay between various PUFAs and their biosynthesis, and not the levels of oleic acid per se, may play a role in modulating fecundity in postdauers adults that experienced starvation.

Next, we investigated how $PD_{Stv}$ animals prioritize fat utilization for reproduction rather than intestinal storage. During *C. elegans* reproduction, intestinal fat stores are reallocated into low-density lipoproteins (LDL)-like particles (yolk lipoproteins or vitellogenins) that are incorporated into oocytes through receptor-mediated endocytosis in a process called vitellogenesis to supply nutrients to the developing embryos (*Kimble and Sharrock, 1983*; *Grant and Hirsh, 1999*). Six vitellogenins homologous to the human ApoB proteins are encoded in the *C. elegans* genome, *vit-1* through *vit-6*, and concomitant multiple RNAi knockdown of the *vit* genes increases adult lifespan in a process that requires NHR-49 and NHR-80 (*Spieth et al., 1991*; *Seah et al., 2016*). Because vitellogenesis mobilizes intestinal fat resources for reproduction and depletes somatic lipid stores (*Kimble and Sharrock, 1983*), we hypothesized that $PD_{Stv}$ adults have reduced fat reservoirs because they prioritize vitellogenesis as a reproductive investment over intestinal storage. To test this, we first examined the lipid content of $PD_{Stv}$ and CON adult embryos in utero using ORO staining. In contrast to the intestinal fat stores, we observed that ORO staining of $PD_{Stv}$ embryos was significantly increased compared to CON embryos (*Figure 4D and E*). Next, we examined the lipid content of CON and $PD_{Stv}$ embryos of *daf-12* mutant strains. If intestinal lipid storage is indeed negatively correlated with the amount of vitellogenesis, we would predict that *daf-12* adults would exhibit the opposite phenotype compared to wild type, with *daf-12* $PD_{Stv}$ embryos having less fat than CON embryos. Instead, we observed that *daf-12* $PD_{Stv}$ and CON embryos have similar levels of ORO staining, all of which were significantly higher than the wild-type levels (*Figure 4D and E*). In contrast to our previous results in *Figure 4B*, these results indicate that DAF-12 does play a role in lipid allocation in control and postdauer animals, potentially by regulating vitellogenesis.

Furthermore, we examined the effects on the intestinal fat stores in control and $PD_{Stv}$ adults when vitellogenesis is disrupted by RNAi knockdown of *vit-1*, which also results in the knockdown of *vit-2/3/4/5* due to the high-sequence homology amongst the *vit* genes (*Seah et al., 2016*; *Figure 5—figure supplement 1*). In animals treated with the empty vector (EV) negative control, $PD_{Stv}$ adults continued to exhibit decreased fat stores compared to control adults. However, $PD_{Stv}$ adults treated with *vit-1* RNAi have significantly greater intestinal fat deposits than $PD_{Stv}$ negative controls (*Figure 5A and B*), indicating that increased vitellogenesis in $PD_{Stv}$ adults may be a contributing factor to the lack of stored intestinal fat. Altogether, these results suggest that $PD_{Stv}$ adults utilize fat accumulated after diapause for reproduction and not somatic storage.

## $PD_{Stv}$ adults exhibit increased longevity

*C. elegans* lifespan is dependent on nutrition and intestinal fat stores. Both animals with increased intestinal fat storage, such as *glp-1* animals, and animals with decreased fat storage, such as dietary restricted animals, exhibit prolonged lifespan (*Kenyon, 2010a*; *Kenyon, 2010b*). To examine

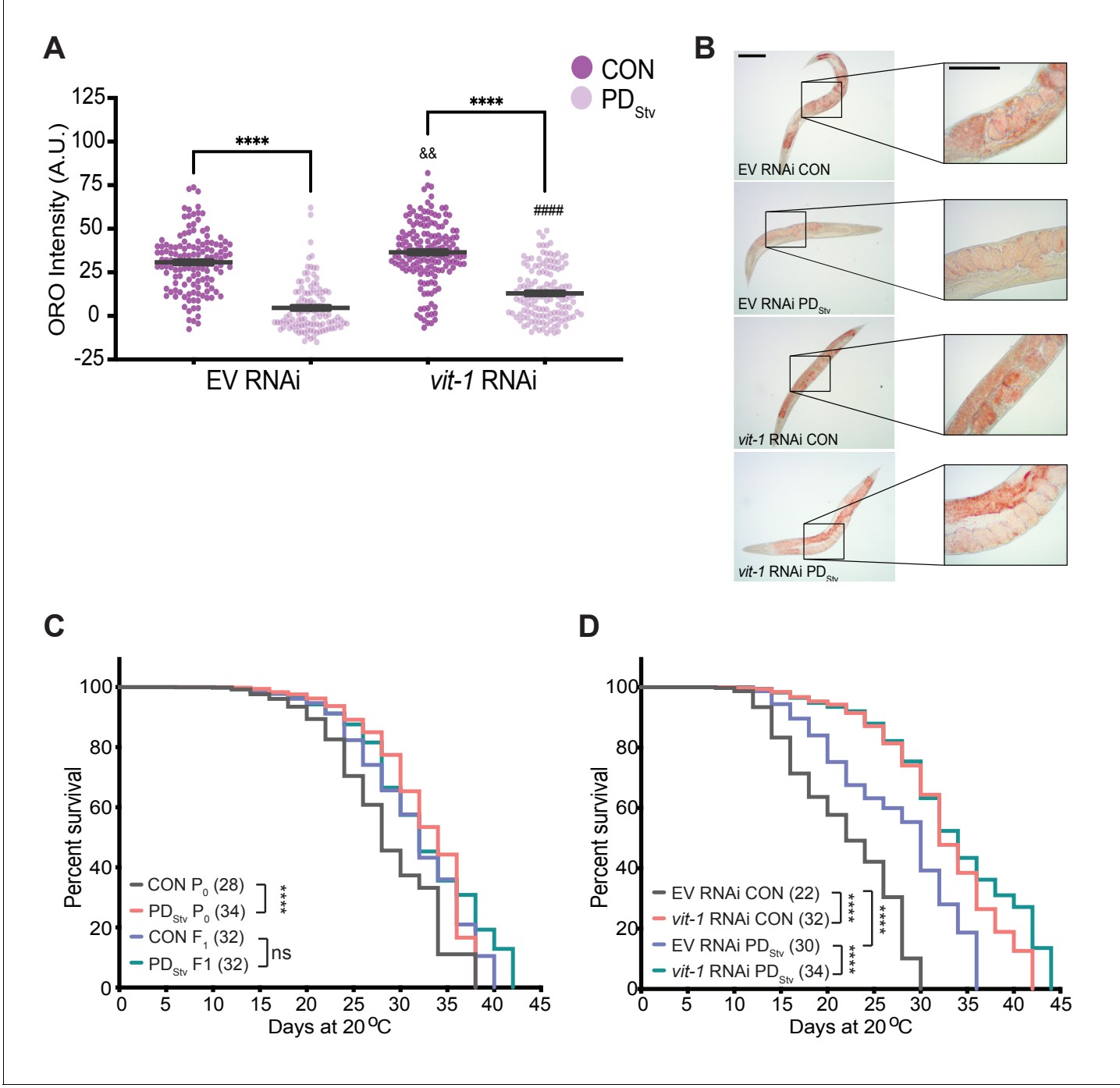

**Figure 5.** Vitellogenesis and adult lifespan are affected in postdauer animals. (A) ORO staining in N2 CON and $PD_{Stv}$ one-day-old adults fed with *vit-1* RNAi or control empty RNAi vector (EV). **** $p < 0.0001$ compares CON and $PD_{Stv}$ of the same RNAi condition; &&$p < 0.01$ and ####$p < 0.0001$ compare *vit-1* to EV RNAi in N2 CON and $PD_{Stv}$, respectively; one-way ANOVA with Sidak's multiple comparison test. Error bars represent S.E.M. Additional data are provided in *Figure 5—source data 1*. (B) Representative micrographs of ORO-stained adults (anterior pharynx as in *Figure 4C*) quantified in (A). Insets show the presence of in utero ORO stained embryos following RNAi knockdown. Scale bars: 100 µM. (C) Adult lifespan assay of N2 CON and $PD_{Stv}$ $P_0$ and $F_1$ generations. **** $p < 0.0001$, ns (not significant); log-rank (Mantel-Cox) test. Median survival (days) is indicated in parenthesis. Additional data are provided in *Figure 5—source data 2*. (D) Adult lifespan assay of N2 CON and $PD_{Stv}$ fed with *vit-1* or control empty vector (EV) RNAi. ****$p < 0.0001$; log-rank (Mantel-Cox) test. Median survival (days) is indicated in parenthesis. Additional data are provided in *Figure 5—source data 3*.

The online version of this article includes the following source data and figure supplement(s) for figure 5:

**Source data 1.** ORO staining in N2 CON and PDStv one-day-adults fed with vit-1 RNAi or control empty RNAi vector (EV).

*Figure 5 continued on next page*

*Figure 5 continued*

**Source data 2.** Adult lifespan assay of N2 CON and PDStv P0 and F1 generations.

**Source data 3.** Adult lifespan assay of N2 CON and PDStv fed with vit-1 or control empty vector RNAi.

**Figure supplement 1.** mRNA level of *vit-1*, *vit-2*, *vit-3/4/5*, and *vit-6* in one-day-old N2 CON and PD$_{Stv}$ adults as measured by reverse transcription quantitative PCR (RT-qPCR) following RNAi-by-feeding with *k09f5.2* (*vit-1*) or control empty RNAi vector.

**Figure supplement 1—source data 1.** mRNA level of vit-1, vit-2, vit-3/4/5, and vit-6 in one-day-old N2 CON and PDStv adults following RNAi-by-feeding with vit-1 or control empty RNAi vector.

**Figure supplement 2.** Altered vitellogenesis does not affect fecundity in postdauer animals.

**Figure supplement 2—source data 1.** Altered vitellogenesis does not affect fecundity in postdauer animals.

**Figure supplement 3.** Reproductive plasticity in postdauer animals is not subject to transgenerational inheritance.

**Figure supplement 3—source data 1.** Reproductive plasticity in postdauer animals is not subject to transgenerational inheritance.

whether the decrease in fat stores in postdauers would affect their longevity, we measured adult lifespan in wild-type controls and postdauers and found that postdauers have a significantly increased longevity compared to controls (*Figure 5C*). Interestingly, we previously reported that crowding-induced postdauers also exhibited increased mean lifespan, suggesting that the increased longevity of PD$_{Stv}$ adults may be due to passage through dauer and not their specific early-life stress (*Hall et al., 2010*). We next asked if a disruption in PD$_{Stv}$ vitellogenesis resulting in increased level of intestinal fat would affect their lifespan by performing *vit-1* knock-down. First, we again observed postdauers fed with EV RNAi lived longer than controls; however, this lifespan differential between controls and postdauers was eliminated when animals were fed with *vit-1* RNAi (*Figure 5D*). Both control and PD$_{Stv}$ animals treated with *vit-1* RNAi exhibited a significant increase in lifespan when compared to cognate animals fed with empty vector (EV) control RNAi, consistent with previous reports of *vit-1* RNAi prolonging lifespan (*Figure 5D*; *Murphy et al., 2003*; *Seah et al., 2016*). However, inhibiting vitellogenesis appears to result in a particular threshold for increased longevity instead of an additive effect, resulting in a similar median lifespan regardless of life history (*Figure 5D*). We next asked whether fecundity was compromised in PD$_{Stv}$ animals as a result of *vit-1* knock-down. Consistent with our previous results, postdauer animals fed with EV RNAi showed a decreased in brood size compared to controls; however, this brood size difference was eliminated following *vit-1* RNAi (*Figure 5—figure supplement 2*). Together, these results support the model that the complex crosstalk between the intestine and germ line shown to regulate somatic aging is also mediating the physiology of postdauer adults (see Discussion).

## Generational transmission of early-life starvation memory

Our results suggest that PD$_{Stv}$ animals upregulate their fatty acid metabolism to increase lipid transport to embryos. In humans, nutritional stress in utero not only promotes metabolic syndrome in adulthood of the affected individuals, but also promotes obesity in subsequent generations (*Painter et al., 2008*; *Veenendaal et al., 2013*). Therefore, we investigated whether subsequent generations inherit the starvation memory by examining if they exhibit any postdauer aging, reproduction, or lipid storage phenotypes. First, we tested if PD$_{Stv}$ F$_1$ progeny had significantly increased longevity compared to CON F$_1$ progeny, but we found no significant differences in lifespan between the two populations (*Figure 5C*). Next, we assessed whether increased fat content in PD$_{Stv}$ embryos affected the reproduction of F$_1$ adults by measuring the brood sizes of CON and PD$_{Stv}$ F$_1$ and F$_2$ progeny, but we found no significant differences beyond the parental generation (*Figure 5—figure supplement 3*). Finally, we examined whether PD$_{Stv}$ progeny exhibited altered lipid content by quantitating intestinal fat storage in control and postdauer F$_1$ and F$_2$ adults using ORO staining. We observed that adult F$_1$ progeny of PD$_{Stv}$ adults had an increased level of stored fat compared to F$_1$ progeny of control adults, but the difference in lipid storage was abolished in the F$_2$ generation (*Figure 6*; *Figure 6—figure supplement 1*). These results indicate that the F$_1$ progeny of PD$_{Stv}$ adults inherit a starvation memory that results in metabolic reprogramming to increase their stored fat reserves.

In *C. elegans*, small RNA pathways often mediate the inheritance of gene expression states (*Feng and Guang, 2013*; *Rechavi and Lev, 2017*). The germline-specific, nuclear Argonaute HRDE-1/WAGO-9 promotes transgenerational silencing via the formation of heterochromatin at targeted genomic loci (*Buckley et al., 2012*; *Feng and Guang, 2013*; *Rechavi and Lev, 2017*). In addition,

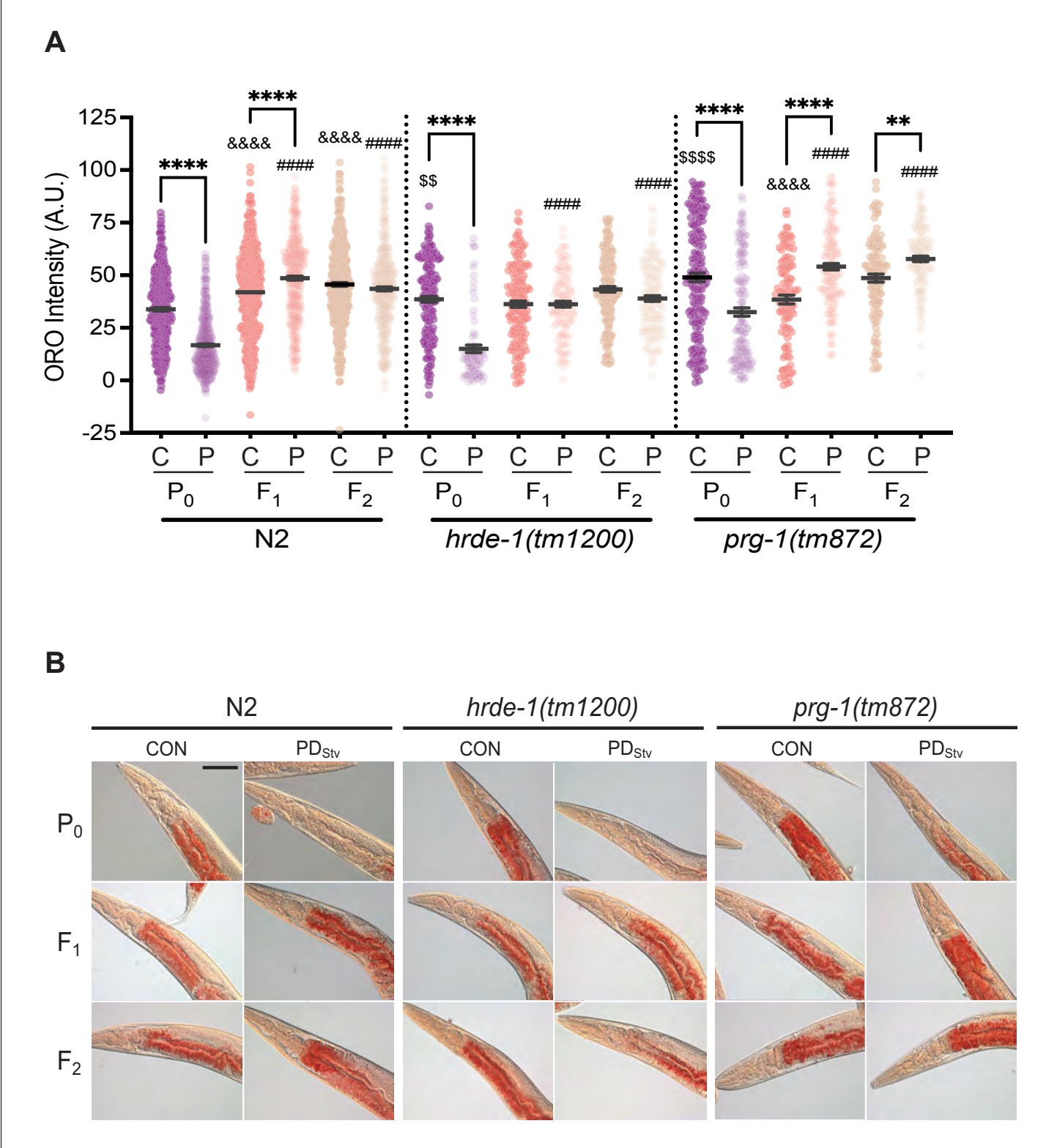

**Figure 6.** Generational inheritance of starvation memory is dependent upon germline-specific RNAi pathways. (**A**) ORO staining in wild-type N2, *hrde-1(tm1200)*, and *prg-1(tm872)* CON and $PD_{Stv}$ one-day-old adults from $P_0$, $F_1$, and $F_2$ generations. *$p < 0.05$ and ****$p < 0.0001$ compares CON (**C**) and $PD_{Stv}$ (**P**) within a genotype and generation; &&&&$p < 0.0001$ compares controls between generations within a strain; ####$p < 0.0001$ compares $PD_{Stv}$ between generations within a strain; $$$p < 0.01$ and $$$$$p < 0.0001$ compares N2 CON to mutant CON of the $P_0$ generation; one-way ANOVA with
*Figure 6 continued on next page*

 Research article

Developmental Biology | Genetics and Genomics

Sidak's multiple comparison test. Error bars represent S.E.M. Additional data are provided in *Figure 6—source data 1*. (B) Representative micrographs of N2, *hrde-1(tm1200)*, and *prg-1(tm872)* CON and PD$_{Stv}$ from P$_0$, F$_1$, and F$_2$ generations stained with ORO quantified in (A). Scale bar: 50 µM.

The online version of this article includes the following source data and figure supplement(s) for figure 6:

**Source data 1.** ORO staining in wild-type N2, hrde-1(tm1200), and prg-1(tm872) CON and PDStv one-day-old adults from P0, F1, and F2 generations.
**Figure supplement 1.** Representative micrographs of N2, *hrde-1(tm1200)*, and *prg-1(tm872)* CON and PD$_{Stv}$ from P$_0$, F$_1$, and F$_2$ generations stained with ORO and quantified in *Figure 6A*.

starvation-induced L1 diapause alters the small RNA populations of subsequent generations in a HRDE-1-dependent manner (*Rechavi et al., 2014*). We therefore asked whether HRDE-1 was required for the generational inheritance of starvation memory by quantifying fat storage in *hrde-1 (tm1200)* control and PD$_{Stv}$ adults and their F$_1$ and F$_2$ progeny. Similar to the wild-type P$_0$ generation, the stored fat levels in *hrde-1* P$_0$ PD$_{Stv}$ adults was significantly reduced when compared to *hrde-1* controls (*Figure 6*; *Figure 6—figure supplement 1*). However, the ORO staining between control and PD$_{Stv}$ adults in both the *hrde-1* F$_1$ and F$_2$ progeny was statistically similar (*Figure 6*; *Figure 6—figure supplement 1*), indicating that HRDE-1 is required for the F$_1$ inheritance of the parental starvation memory that promotes increased lipid storage in wild-type animals.

To further investigate the role of the generational inheritance of a starvation memory, we examined the effect of the *prg-1(tm872)* mutation on the fat stores of control and PD$_{Stv}$ adults and their F$_1$ and F$_2$ progeny. PRG-1 is a Piwi-class Argonaute that acts upstream of HRDE-1 to perpetuate transgenerational epigenetic memory in the germ line (*Ashe et al., 2012*; *Shirayama et al., 2012*). We found that in the P$_0$ generation, PD$_{Stv}$ *prg-1(tm872)* adults exhibited a significant decrease in stored fats compared to controls, similar to wild type (*Figure 6*; *Figure 6—figure supplement 1*). Also similar to wild type, the F$_1$ progeny of *prg-1* PD$_{Stv}$ animals exhibited increased intestinal lipid storage. However, the F$_2$ progeny of CON and PD$_{Stv}$ *prg-1* mutants continued to exhibit the increased PD$_{Stv}$ fat stores phenotype instead of 'resetting' like in the wild type (*Figure 6*; *Figure 6— figure supplement 1*), suggesting that PRG-1 plays a role in erasing the starvation memory inherited from PD$_{Stv}$ adults in the F$_2$ generation. Although PRG-1 and HRDE-1 work in the same nuclear RNAi pathway required for transgenerational inheritance, our ORO staining show that these proteins play different roles in the transmission of an ancestral starvation memory. Namely, HRDE-1 promotes the inheritance of the starvation memory to the next generation, and PRG-1 halts the propagation of an 'expired' memory to the grand-progeny. In addition, the P$_0$ CON of adults of *hrde-1* and *prg-1* mutant strains have significantly increased stored fat compared to wild type, indicating these pathways also seem to play a role in regulating lipid storage in continuously developing animals (*Figure 6*; *Figure 6—figure supplement 1*; *Figure 7—figure supplement 2*). Altogether, our results show that *C. elegans*, like humans, inherit the disposition for increased adiposity from parents that experienced early-life starvation.

## Steroid signaling, reproductive longevity, and fatty acid metabolic pathways act synergistically at different developmental time points to regulate reproductive plasticity

Thus far, our results indicate that the fatty acid metabolism and reproductive longevity pathways are required for the reduced fecundity phenotype in PD$_{Stv}$ adults. While our data suggests a role for DA-dependent DAF-12 activity in regulating vitellogenesis, its contribution to regulating the reduced fecundity of postdauer adults is less certain given the multiple possible interpretations of our results. Moreover, how these pathways are potentially interacting to regulate PD$_{Stv}$ phenotypes is unclear. The *daf-12*, *nhr-49*, and *tcer-1* PD$_{Stv}$ mutant phenotypes are distinct, suggesting the possibility that they may act in different pathways, tissues, or developmental time points to regulate PD$_{Stv}$ fecundity. Furthermore, strains carrying double mutations in *daf-12* and either *tcer-1* or *kri-1* have control and postdauer brood sizes similar to the *daf-12* mutations alone, suggesting that *daf-12* alleles are acting downstream and masking the phenotypes of the reproductive longevity mutants (Appendix 1; *Figure 1—figure supplement 1B*).

To further investigate the developmental mechanism of steroid signaling, reproductive longevity, and fatty acid metabolism pathways in the regulation of reproductive plasticity, we examined if

DAF-12, TCER-1, and NHR-49 play a direct role in the timing of germline proliferation in postdauer larvae. We previously demonstrated that wild-type PD$_{Stv}$ animals delay the onset of germline proliferation compared to control animals, contributing to a reduction in brood size (*Ow et al., 2018*). In *C. elegans* hermaphrodites, undifferentiated germ cells initiate spermatogenesis during the L4 larval stage, followed by a transition to oogenesis at the adult stage (*L'Hernault, 2006*). Therefore, reproduction in *C. elegans* hermaphrodites is sperm-limited (*Byerly et al., 1976*; *Ward and Carrel, 1979*; *Kimble and Ward, 1988*). In previous results, when control and PD$_{Stv}$ somatic development was synchronized using vulva morphology, we observed significantly fewer germ cell rows in PD$_{Stv}$ larvae compared to control larvae, correlating with fewer sperm available for self-fertilization in adulthood (*Ow et al., 2018*). To determine if DAF-12, TCER-1, and NHR-49 play a direct role in germline proliferation as a mechanism to regulate reproductive plasticity, we counted the number of germ cell rows in control and PD$_{Stv}$ mutant larvae that were developmentally synchronized by their somatic morphology. Because *daf-12(rh284)* and *daf-12(rh285)* mutants have altered gonad morphologies that prevent accurate synchronization, we used *daf-36(k114)* to disrupt the steroid signaling pathway. First, we recapitulated our previous results showing that wild-type PD$_{Stv}$ larvae have significantly fewer total germ cell rows compared to control larvae due to significantly reduced cell rows in

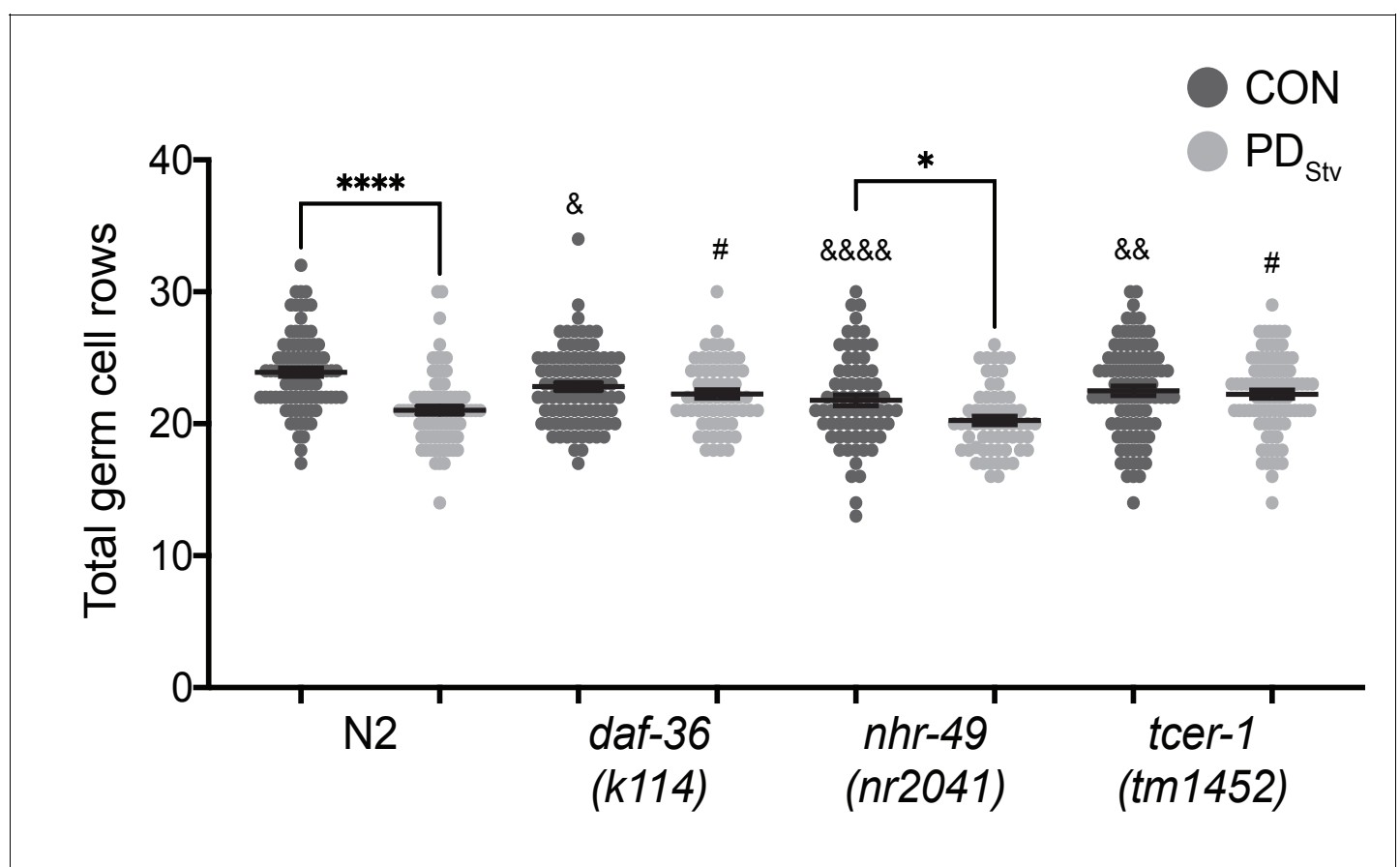

**Figure 7.** DAF-36 and TCER-1 regulate the onset of germline proliferation. Total germ cell rows in CON and PD$_{Stv}$ wild-type N2 and mutant larva exhibiting L3 vulva morphology (see Materials and Methods). * $p < 0.05$ and **** $p < 0.0001$ compare CON and PD$_{Stv}$ within a genotype; [&] $p < 0.05$, [&&]$p < 0.01$, and [&&&&]$p < 0.0001$ compare N2 CON with mutant CON of mutants; [#]$p < 0.05$ compares PD$_{Stv}$ of N2 to PD$_{Stv}$ of mutants; one-way ANOVA with Sidak's multiple comparison test. Error bars represent S.E.M. Additional data are provided in *Figure 7—source data 1*.

The online version of this article includes the following source data and figure supplement(s) for figure 7:

**Source data 1.** Total germ cell rows in CON and PDStv wild-type N2 and mutant larva.

**Figure supplement 1.** DAF-36 and TCER-1 are required for the delay in the onset of germline proliferation in postdauer animals.

**Figure supplement 1—source data 1.** DAF-36 and TCER-1 are required for the delay in the onset of germline proliferation in postdauer animals.

**Figure supplement 2.** Model of endocrine signaling pathways acting as contributors to fatty acid metabolism of PD$_{Stv}$ animals.

the meiotic transition zone (*Figure 7*; *Figure 7—figure supplement 1*). In contrast, *daf-36(k114)* control and PD$_{Stv}$ larvae did not exhibit a significant difference in total, mitotic, or meiotic cell rows, indicating that DAF-36-dependent DA is required during early germline development for the delay in PD$_{Stv}$ germ cell proliferation (*Figure 7*; *Figure 7—figure supplement 1*). This result is consistent with the increased brood size of PD$_{Stv}$ adults in *daf-12(rh284)* and *daf-12(rh285)* mutants, which express DAF-12 proteins unable to bind to DA.

In contrast to the *daf-36* mutant, the *nhr-49(nr2041)* mutant exhibited a significant decrease in the number of total germ cell rows in PD$_{Stv}$ larvae relative to control larvae, similar to wild-type animals (*Figure 7*), although the numbers of germ cell rows in the mitotic and meiotic zones of *nhr-49* control and PD$_{Stv}$ animals were not significantly different (*Figure 7—figure supplement 1*). These results suggest that NHR-49 acts later in development to regulate the PD$_{Stv}$ brood size, perhaps by modulating fatty acid metabolism in the intestine to support vitellogenesis in adults. Like NHR-49, TCER-1 plays an intestinal role in upregulating fatty acid metabolism genes in animals lacking germ cells, but also functions redundantly with PUF-8, a member of an evolutionarily conserved stem cell proliferation modulatory family, to potentiate germ cell proliferation in animals undergoing continuous development (*Pushpa et al., 2013*; *Amrit et al., 2016*). We found that a mutation in *tcer-1* resulted in a similar number of total germ cell rows in control and PD$_{Stv}$ larvae, indicating that TCER-1 plays a role in PD$_{Stv}$ reproductive plasticity by regulating germ cell proliferation in addition to its intestinal role (*Figure 7*). Interestingly, *tcer-1* PD$_{Stv}$ larvae also appear to have a defect in the onset of meiosis, since PD$_{Stv}$ larvae have significantly greater number of mitotic germ cell rows compared to controls (*Figure 7—figure supplement 1*). Together, these results indicate that DAF-12 and TCER-1 act early in germline development to delay the onset of germ cell proliferation in PD$_{Stv}$ animals, while NHR-49 primarily functions later in development to promote the reduced fertility of PD$_{Stv}$ adults.

## Discussion

The trade-off between reproduction and longevity has long been associated with the notion that in the absence of reproduction, the fat stores of an animal would be redistributed to promote somatic maintenance (*Williams, 1966*; *Kirkwood, 1977*; *Westendorp and Kirkwood, 1998*; *Partridge et al., 2005*). As such, reducing or suppressing reproduction in animals often extends lifespan, in part due to enhanced lipogenic processes (*Fowler and Partridge, 1989*; *Kenyon, 2010a*; *Kenyon, 2010b*; *Judd et al., 2011*). What remains unclear is how reproduction and somatic maintenance are balanced by developmental signals. In this study, we show reproduction is attenuated in wild-type *C. elegans* that have experienced dauer as a result of early-life starvation. We propose a model whereby steroid signaling, reproductive longevity, and fatty acid metabolic pathways are reprogrammed in animals that experienced starvation-induced dauer in order to delay the onset of germline proliferation and redistribute intestinal fat to developing oocytes (*Figure 7—figure supplement 2*). These changes would allow PD$_{Stv}$ adults to delay reproduction until energy thresholds are met to provide adequate levels of nutrition to fewer albeit viable embryos at the expense of somatic survival and maintenance. In *glp-1* mutants that lack a germ line, the fat that would normally be allocated to the progeny is channeled to nurture somatic tissues and, consequently, promote extended longevity. Thus, the mechanisms that prolong lifespan in the absence of a functional germ line are the same cellular programs deployed to respond to developmental signals triggered by early-life starvation in a wild-type animal.

The extended lifespan associated with animals missing a functional germ line is specifically dependent on the lack of proliferating germ cells and not due to sterility resulting from sperm, oocyte, or meiotic precursor cell deficiency (*Hsin and Kenyon, 1999*; *Arantes-Oliveira et al., 2002*). Animals without a functional germ line have upregulated fat metabolism pathways and exhibit increased levels of intestinal fat stores that are associated with a longer lifespan (*Kenyon, 2010a*; *Kenyon, 2010b*; *Amrit et al., 2016*). However, a direct correlation between an extended adult lifespan and an increase in intestinal lipid level remains to be elucidated. Our results demonstrated that postdauers with lower somatic fat content also exhibited longer lifespans (*Figures 4B, C* and *5C*), indicating that an increased intestinal fat stores are not required for lifespan extension. Indeed, dietary restriction in animals, whether chronic or intermittent, can promote longevity through multiple different mechanisms (*Kenyon, 2010b*). Although PD$_{Stv}$ adults have transcriptional signatures

similar to *glp-1* mutants, their phenotype is more similar to *eat-2* mutants, which are a genetic model for chronic dietary restriction due to a pharyngeal pumping defect (*Avery, 1993*). *Eat-2* mutants have significantly decreased ORO staining and a lifespan increase up to 50% over wild type (*Lakowski and Hekimi, 1998*; *Klapper et al., 2011*). In addition, *eat-2* mutants delay reproduction and have a significantly smaller brood size compared to wild type (*Crawford et al., 2007*). The lifespan extension of *eat-2* requires TOR inhibition through PHA-4 as well as the activity of SKN-1 (*Bishop and Guarente, 2007*; *Sheaffer et al., 2008*), but the expression of the genes encoding these proteins are unchanged in PD$_{Stv}$ adults. However, passage through dauer, a nonfeeding stage, may trigger an independent mechanism of dietary restriction through somatic aging pathways that regulates the PD$_{Stv}$ phenotypes.

The cellular mechanisms that regulate reproduction are intricately connected to lipid metabolism and longevity (*Wang et al., 2008*). While a number of genes and cellular components affecting germline proliferation have been extensively investigated (*Kimble and Crittenden, 2005*), what might the actual signals communicating the state of proliferating germ cells be that arbitrate lipid levels and the aging process? Here, we argue the signal communicating the state of germ cell proliferation may include dafachronic acids. Dafachronic acids mediating increased longevity are produced in the somatic gonad, which includes the stem cell niche site of the germ line (*Yamawaki et al., 2010*). Based on our results, we conclude that DA is required for the delay in germline proliferation in PD$_{Stv}$ animals (*Figure 7*; *Figure 7—figure supplement 1*), which is consistent with previous data indicating that DA can inhibit germ cell proliferation in adults in a DAF-12 dependent manner (*Mukherjee et al., 2017*). Our data also suggests that DAF-12 may be acting at multiple developmental stages and in different tissue types to regulate PD$_{Stv}$ reproduction. Our germ cell row counts indicate that *daf-36* mutant larvae have no significant defects at the onset of germline proliferation that would account for the low brood size observed in control adults (*Figure 7*). However, *daf-12*, *daf-36*, and *daf-9* mutants have severe gonad defects, including distal tip migration defects, that can impair adult reproduction (*Antebi et al., 2000*; *Gerisch et al., 2001*; *Rottiers et al., 2006*). Our brood size data alone is consistent with the hypothesis that the gonad defects may be partially rescued after passage through dauer resulting in an increased brood size compared to controls, or that DA and DAF-12 are not required to regulate postdauer brood size (*Figure 1B*). However, we demonstrated that the *daf-12 rh284* and *rh285* alleles can mask the phenotype of reproductive longevity pathway mutants, indicating that DAF-12, but not DA, is required for the PD$_{Stv}$ brood size phenotype (*Figure 1—figure supplement 1B*). One possible mechanism of how DAF-12 activity contributes to the reproductive plasticity in PD$_{Stv}$ adults through its regulation of vitellogenesis. We showed that the levels of stored lipids in *daf-12* embryos were higher than that observed in embryos of either control or PD$_{Stv}$ wild-type adults and was independent of the levels of intestinal lipid storage (*Figure 4*). Interestingly, the connections between steroid hormone signaling, vitellogenesis, and fertility are well documented in various fish species (e.g. *King et al., 2003*; *Wu et al., 2021*), and exposing *C. elegans* to exogenous cholesterol, the precursor for DA, has been shown to increase expression of vitellogenesis genes (*Novillo et al., 2005*). Thus, we are currently investigating the mechanisms of how DAF-12 and DA regulate lipid homeostasis and vitellogenesis to modulate reproduction in PD$_{Stv}$ animals. Taken together, our data are consistent with a model where DA and DAF-12 signaling act tissue specifically to regulate germline proliferation and vitellogenesis based on the life history of the animal (*Figure 7—figure supplement 2*).

An intriguing finding of our study is that the parental starvation memory of PD$_{Stv}$ adults was bequeathed to the F$_1$ progeny in a HRDE-1 dependent manner, triggering elevated levels of fat stores, presumably as a physiological defense against future famine (*Figure 6*; *Figure 6—figure supplement 1*). In the wild-type grand-progenies, PRG-1 is required for the increase in fat stores to be reset to control levels of the same generation. One potential explanation is that small RNA signals are transmitted to subsequent generations via the HRDE-1 and/or PRG-1 RNAi pathways to effect somatic phenotypes. However, with the exception of DAF-16, none of the germline longevity pathway genes or the vitellogenesis genes examined in this study were categorized as direct HRDE-1 targets (*Buckley et al., 2012*). Given that the life stage (adulthood) at which the HRDE-1 targets were identified is the same life stage that was used in this study, we speculate that HRDE-1 may be indirectly targeting endocrine and vitellogenins genes by: (1) targeting germ line genes that then affect somatic gene expression or (2) indirectly regulating the function of the endocrine and vitellogenin genes by targeting a different repertoire of somatic targets. Interestingly, we find that 62% of small

RNAs associated with HRDE-1 target genes (984 out of 1587) are expressed in somatic tissues, such as neurons, intestine, hypodermis, and muscle (*Ortiz et al., 2014*; *Kaletsky et al., 2018*). Accordingly, HRDE-1 is known to contribute to the heritability of a cohort of small RNAs targeting nutrition and lipid transporter genes that was inherited for at least three generations from populations that experienced L1 larval arrest (*Rechavi et al., 2014*). In addition, HRDE-1 is required for the repression of a group of genes activated upon multi-generational high-temperature stress that is inherited for at least two generations in the absence of the stress (*Ni et al., 2016*). Similarly, PRG-1 has been reported to function in somatic tissue by repressing *C. elegans* axonal regeneration (*Lee et al., 2012*; *Shen et al., 2018*; *Kim et al., 2018*), and reports from *Drosophila*, mollusks, and mammals have shown that piRNAs are expressed in the nervous system (*Lee et al., 2011*; *Rajasethupathy et al., 2012*; *Perrat et al., 2013*; *Nandi et al., 2016*). Recently, PRG-1 was shown to potentiate the transgenerational inheritance of learned avoidance to the pathogenic PA14 *Pseudomonas aeruginosa* bacteria for multiple generations (*Moore et al., 2019*). Thus, it is likely that HRDE-1 and PRG-1 RNAi pathways may serve as signaling referees between the soma and the germ line to effect changes due to environmental and developmental signals to perdure ancestral starvation memory.

Our study shows that PD<sub>Stv</sub> adults have upregulated expression of lipid metabolism genes as a means to load embryos with increased fat and potentially protect progeny against the consequences of future food scarcity. During the course of its natural history, *C. elegans* occupies ephemeral environments such as rotting fruit or decomposing vegetation, where conditions and food availability are highly unpredictable. The dauer larva affords *C. elegans* a survival and dispersal strategy to escape harsh environmental conditions by often associating with passing invertebrate carriers. Once a food source is found, dauers resume reproductive development to colonize the new habitat. Upon exhaustion of resources and population expansion, young larvae enter dauer and thereby repeating the 'boom and bust' life cycle (*Schulenburg and Félix, 2017*). Because of frequent environmental perturbations, an adopted phenotypic plasticity strategy would ensure an advantage in species survival. The generation following a bust period would inherit the cellular programs for increased somatic lipid stores. It is thus remarkable that the cellular mechanisms to ensure survival of the species are fundamentally similar between humans and nematodes, two species that have diverged hundreds of millions of years ago, once again underscoring the relevance of a simple roundworm in understanding basic animal physiology.

# Materials and methods

**Key resources table**

| Reagent type (species) or resource | Designation | Source or reference | Identifiers | Additional information |
|---|---|---|---|---|
| Gene (include species here) | | | | |
| Strain, strain background (*Caenorhabditis elegans*) | N2 | Caenorhabditis Genetics Center | Wild type | |
| Strain, strain background (*Caenorhabditis elegans*) | AA82 | Caenorhabditis Genetics Center | *daf-12(rh284) X* | |
| Strain, strain background (*Caenorhabditis elegans*) | AA85 | Caenorhabditis Genetics Center | *daf-12(rh285) X* | |
| Strain, strain background (*Caenorhabditis elegans*) | AA292 | Caenorhabditis Genetics Center | *daf-36(k114) V* | |
| Strain, strain background (*Caenorhabditis elegans*) | AA1052 | Adam Antebi, Max Planck Institute | *dhs-16(tm1890) V* | |

*Continued on next page*

*Continued*

| Reagent type (species) or resource | Designation | Source or reference | Identifiers | Additional information |
|---|---|---|---|---|
| Strain, strain background (*Caenorhabditis elegans*) | AE501 | Caenorhabditis Genetics Center | *nhr-8(ok186) IV* | |
| Strain, strain background (*Caenorhabditis elegans*) | BS1080 | Tim Schedl, Washington University | *gld-1::gfp::3xflag* | |
| Strain, strain background (*Caenorhabditis elegans*) | BX26 | Caenorhabditis Genetics Center | *fat-2(wa17) IV* | |
| Strain, strain background (*Caenorhabditis elegans*) | BX106 | Caenorhabditis Genetics Center | *fat-6(tm331) IV* | |
| Strain, strain background (*Caenorhabditis elegans*) | BX107 | Caenorhabditis Genetics Center | *fat-5(tm420) V* | |
| Strain, strain background (*Caenorhabditis elegans*) | BX110 | Caenorhabditis Genetics Center | *fat-6(tm331) IV; fat-5(tm420) V* | |
| Strain, strain background (*Caenorhabditis elegans*) | BX153 | Caenorhabditis Genetics Center | *fat-7(wa36) V* | |
| Strain, strain background (*Caenorhabditis elegans*) | BX156 | Caenorhabditis Genetics Center | *fat-6(tm331) IV; fat-7(wa36) V* | |
| Strain, strain background (*Caenorhabditis elegans*) | BX160 | Caenorhabditis Genetics Center | *fat-7(wa36) fat-5(tm420) V* | |
| Strain, strain background (*Caenorhabditis elegans*) | CB1375 | Caenorhabditis Genetics Center | *daf-18(e1375) IV* | |
| Strain, strain background (*Caenorhabditis elegans*) | CE541 | Caenorhabditis Genetics Center | *sbp-1(ep79) III* | |
| Strain, strain background (*Caenorhabditis elegans*) | CF1139 | Caenorhabditis Genetics Center | *daf-16(mu86) I; muIs61 [(pKL78) daf16::gfp + rol-6(su1006)]* | |
| Strain, strain background (*Caenorhabditis elegans*) | CF2052 | Caenorhabditis Genetics Center | *kri-1(ok1251) I* | |
| Strain, strain background (*Caenorhabditis elegans*) | CF2167 | Caenorhabditis Genetics Center | *tcer-1(tm1452) II* | |
| Strain, strain background (*Caenorhabditis elegans*) | EG6699 | Caenorhabditis Genetics Center | *ttTi5605 II; unc-119(ed3) III; oxEx1578 [eft-3p::gfp + Cbr-unc-119]* | |
| Strain, strain background (*Caenorhabditis elegans*) | GR2063 | Caenorhabditis Genetics Center | *hsd-1(mg433) I* | |
| Strain, strain background (*Caenorhabditis elegans*) | RG1228 | Caenorhabditis Genetics Center | *daf-9(rh50) X* | |
| Strain, strain background (*Caenorhabditis elegans*) | SEH301 | This study | *nhr-13(gk796) V backcrossed* | Sarah Hall, Syracuse University |
| Strain, strain background (*Caenorhabditis elegans*) | SEH302 | This study | *nhr-49(nr2041) I; nhr-80(tm1011) III* | Sarah Hall, Syracuse University |

*Continued on next page*

Continued

| Reagent type (species) or resource | Designation | Source or reference | Identifiers | Additional information |
|---|---|---|---|---|
| Strain, strain background (*Caenorhabditis elegans*) | SEH303 | This study | *nhr-49(nr2041) I; nhr-13(gk796) V* | Sarah Hall, Syracuse University |
| Strain, strain background (*Caenorhabditis elegans*) | SEH304 | This study | *nhr-49(nr2041) I; nhr-80(tm1011) III; nhr-13(gk796) V* | Sarah Hall, Syracuse University |
| Strain, strain background (*Caenorhabditis elegans*) | SEH312 | This study | *daf-16(mu86) I; muEx158 (daf-16^AM::gfp + sur-5p::gfp)* | Sarah Hall, Syracuse University |
| Strain, strain background (*Caenorhabditis elegans*) | SEH319 | This study | *nhr-49(et8) I backcrossed* | Sarah Hall, Syracuse University |
| Strain, strain background (*Caenorhabditis elegans*) | SEH326 | This study | *nhr-49(et13) I backcrossed* | Sarah Hall, Syracuse University |
| Strain, strain background (*Caenorhabditis elegans*) | SEH327 | This study | *nhr-49(et7) I backcrossed* | Sarah Hall, Syracuse University |
| Strain, strain background (*Caenorhabditis elegans*) | SEH342 | This study | *nhr-49(nr2041) I; nhr-66(ok940) IV* | Sarah Hall, Syracuse University |
| Strain, strain background (*Caenorhabditis elegans*) | SEH343 | This study | *nhr-49(gk405) I backcrossed* | Sarah Hall, Syracuse University |
| Strain, strain background (*Caenorhabditis elegans*) | SEH344 | This study | *nhr-49(ok2165) I backcrossed* | Sarah Hall, Syracuse University |
| Strain, strain background (*Caenorhabditis elegans*) | SEH350 | This study | *pqm-1(ok485) II backcrossed* | Sarah Hall, Syracuse University |
| Strain, strain background (*Caenorhabditis elegans*) | SEH351 | This study | *kri-1(ok1251) I; daf-12(rh284) X* | Sarah Hall, Syracuse University |
| Strain, strain background (*Caenorhabditis elegans*) | SEH352 | This study | *kri-1(ok1251) I; daf-12(rh285) X* | Sarah Hall, Syracuse University |
| Strain, strain background (*Caenorhabditis elegans*) | SEH353 | This study | *tcer-1(tm1452) II; daf-12(rh284) X* | Sarah Hall, Syracuse University |
| Strain, strain background (*Caenorhabditis elegans*) | SEH354 | This study | *tcer-1(tm1452) II; daf-12(rh285) X* | Sarah Hall, Syracuse University |
| Strain, strain background (*Caenorhabditis elegans*) | SEH357 | This study | *glp-4(bn2) I; pdrSi1 [Pglp-4::glp-4 cDNA::gfp ::glp-4 3'UTR; unc-119(+)] II* | Sarah Hall, Syracuse University |
| Strain, strain background (*Caenorhabditis elegans*) | SEH368 | This study | *glp-4(bn2) I; pdrSi2 [Pnhx-2::glp-4 cDNA::gfp: :glp-4 3'UTR; unc-119(+)] II* | Sarah Hall, Syracuse University |
| Strain, strain background (*Caenorhabditis elegans*) | SEH369 | This study | *glp-4(bn2) I; pdrSi3 [Pzfp-2::glp-4 cDNA::gfp::glp-4 3'UTR; unc-119(+)] II* | Sarah Hall, Syracuse University |
| Strain, strain background (*Caenorhabditis elegans*) | SEH370 | This study | *glp-4(bn2) I; pdrSi4 [Ppgl-1::glp-4 cDNA::gfp ::glp-4 3'UTR; unc-119(+)] II* | Sarah Hall, Syracuse University |
| Strain, strain background (*Caenorhabditis elegans*) | SEH383 | This study | *hrde-1(tm1200) III backcrossed* | Sarah Hall, Syracuse University |

*Continued on next page*

*Continued*

| Reagent type (species) or resource | Designation | Source or reference | Identifiers | Additional information |
|---|---|---|---|---|
| Strain, strain background (*Caenorhabditis elegans*) | SS104 | Caenorhabditis Genetics Center | *glp-4(bn2) I* | |
| Strain, strain background (*Caenorhabditis elegans*) | SP488 | Caenorhabditis Genetics Center | *smk-1(mn156) V* | |
| Strain, strain background (*Caenorhabditis elegans*) | STE68 | Caenorhabditis Genetics Center | *nhr-49(nr2041) I* | |
| Strain, strain background (*Caenorhabditis elegans*) | STE69 | Caenorhabditis Genetics Center | *nhr-66(ok940) IV* | |
| Strain, strain background (*Caenorhabditis elegans*) | STE70 | Caenorhabditis Genetics Center | *nhr-80(tm1011) III* | |
| Strain, strain background (*Caenorhabditis elegans*) | STE73 | Caenorhabditis Genetics Center | *nhr-80(tm1011) III; nhr-13(gk796) V* | |
| Strain, strain background (*Caenorhabditis elegans*) | TJ356 | Caenorhabditis Genetics Center | *zIs356 [daf-16p::daf-16a/b::gfp + rol-6(su1006)]* | |
| Strain, strain background (*Caenorhabditis elegans*) | WM161 | Caenorhabditis Genetics Center | *prg-1(tm872) II* | |
| Strain, strain background (*Caenorhabditis elegans*) | XA7702 | Caenorhabditis Genetics Center | *mdt-15(tm2182) III* | |
| Strain, strain background (*Escherichia coli*) | OP50 | Caenorhabditis Genetics Center | OP50 | |
| Recombinant DNA reagent | *k09f5.2* | **Kamath et al., 2001** | RNAi | |
| Chemical compound, drug | Oleic acid C18:1 | NuChek Prep, Inc.; Elysian, Minnesota | Cat no. U-46-A | |
| Chemical compound, drug | Oil Red O (ORO) | Sigma Aldrich | Cat no. O0625 | |
| Chemical compound, drug | 5-fluoro-2'-deoxyuridine (FUDR) | Sigma Aldrich | Cat no. F0503 | |
| Chemical compound, drug | IPTG | Sigma Aldrich | Cat no. I5502 | |
| Chemical compound, drug | Carbenicillin | Sigma Aldrich | Cat no. C1389 | |
| Chemical compound, drug | $\Delta^7$ form of dafachronic acid | Frank Schroeder, Cornell University | | |
| Chemical compound, drug | DAPI stain | Thermo Scientific | | Used at a concentration of 1:1000 |
| Software, algorithm | Spot 5.2 | Nikon | Nikon Eclipse | |
| Software, algorithm | GraphPad Prism | GraphPad Software | v.9 | |
| Software, algorithm | ImageJ software | ImageJ (http://imagej.nih.gov/ij/) | | |

## C. elegans **strains and husbandry**

N2 Bristol wild-type strain was used as the reference strain. Worms were grown at 20°C unless otherwise indicated in Nematode Growth Medium (NGM) seeded with *Escherichia coli* OP50 using

standard methods (*Brenner, 1974*; *Stiernagle, 2006*). Mutants that were not previously backcrossed were backcrossed at least four times to our laboratory N2 wild type before use. Control and starvation-induced postdauer animals were obtain in a similar manner as described before (*Ow et al., 2018*). Briefly, to gather PD$_{Stv}$ animals, well-fed worms grown on seeded NGM plates and monitored until the bacteria food was depleted and dauers were visible (about 1 week). Dauers were selected with 1% SDS, followed by recovery by feeding on seeded NGM plates. One-day-old PD$_{Stv}$ adults were collected on day two following recovery (first day of adulthood). Control adults were obtained by collecting embryos from hypochlorite-treated well-fed gravid adults that did not experience dauer. Embryos were grown on seeded NGM plates until the first day of adulthood. All strains used in this study are listed in *Supplementary file 2*.

## Brood assays

Ten L4 larvae were placed individually onto 35 mm NGM plates seeded with *E. coli* OP50 and incubated at 20℃. Animals were transferred daily to fresh 35 mm NGM plates until egg laying ceased. Live progeny from each egg laying plate were counted. Assays were performed from at least three biological independent replicates.

## Oleic acid (OA) supplementation

Animals were induced into dauer by starvation as well as recovered on peptone-less NGM plates seeded with *E. coli* OP50 pre-loaded with oleic acid (NuChek Prep, Inc.; Elysian, Minnesota) as described by *Devkota et al., 2017*. OP50 was grown overnight at 37℃ in liquid LB supplemented with 600 μM of oleic acid or with an equivalent volume of ethanol (the oleic acid solvent) to serve as the control. Cultures were pelleted and washed several times with M9 buffer (*Stiernagle, 2006*) and resuspended at a 10x concentration. The 10x OP50 was seeded onto peptone-less NGM plates and allowed to dry overnight before use. At least three independent replicates were performed.

## Oil Red O (ORO) staining

Fat stores were stained using ORO dye as described by *O'Rourke et al., 2009*. Age matched one-day-old adults were washed from 60 mm seeded NGM plates with 1x PBS pH 7.4 and rinsed 3–4 times until they were cleared of bacteria. Worms were permeabilized in 1x PBS pH 7.4 with an equal volume of 2x MRWB buffer (160 mM KCl, 40 mM NaCl, 14 mM Na$_2$EGTA, 1 mM spermidine-HCl, 0.4 mM spermine, 30 mM PIPES pH 7.4, 0.2% β-mercaptoethanol) and supplemented with 2% paraformaldehyde. Samples were rocked for 1 hr at room temperature. Following fixation, worm samples were washed with 1x PBS pH 7.4, resuspended in 60% isopropanol, and incubated at room temperature for 15 min. An ORO stock solution (prepared beforehand as a 0.5 g/100 mL in isopropanol and equilibrated for several days) was diluted to 60% with dH$_2$O and rocked for at least one day to be used as the working stock. The ORO working stock was filtered through a 0.22 or 0.45 μm filter immediately before use. Fixed worms were incubated in filtered ORO working stock and rocked overnight at room temperature. Next day, worm samples were allowed to settle and the ORO stain was removed. Worm pellets were washed once with 1x PBS pH 7.4 and resuspended in 200 μL of 1x PBS with 0.01% Triton X-100. Aliquots of worm samples were mounted onto microscope glass slides and imaged. Quantification of embryo ORO staining was done by singling out individual embryos in utero from one-day-old adults. Images were captured with a Nikon Eclipse Ci with Spot 5.2 software, an iPhone through iDu Optics equipped with a LabCam adapter (New York, NY), or with a Leica DM5500 B microscope with the LAS X Core Workstation fitted with a MC170 Color HD camera. All images from parallel experiments were captured using the same microscope platform. Color images were separated into their RGB channel components and the intensity of staining in the anterior intestine was measured on the green channel as previously described (*Yen et al., 2010*) using ImageJ (NIH). Because the unstained pharynx immediately above the anterior intestine was used as the ORO staining subtraction background, negative ORO staining values (*Figures 5A* and *6A*) are a result of a high background in specific worm samples.

## Fatty acid analysis

To obtain control animals, small agar chunks (approximately 1 cm x 1 cm) from a well-fed mixed population of worms grown on NGM plates at 20℃ were transferred to 100 mm NGM plates freshly

seeded with 10x concentrated OP50 (10x NGM plates). After 3 days of propagation, embryos were harvested by standard methods using sodium hypochlorite (*Stiernagle, 2006*) and transferred to 10x NGM plates. One-day-old adults were collected three days later and washed with Milli-Q water until the supernatant was clear. Excess water was removed by centrifugation (3000 rpm for 30 s) and worm pellets (0.25 to 1.09 g) were flash frozen in a dry ice and ethanol slurry and stored at −80°C until analysis. To obtain postdauer animals, agar chunks from worms grown in a similar manner as control animals were transferred to 10x NGM plates and incubated for 2 weeks at 20°C for starvation-induced dauer formation. Starved worms were collected with Milli-Q water and dauers were selected by treatment with 1% SDS (*Stiernagle, 2006*). Dauers were transferred to 10x NGM plates and fed for 2 days. Postdauer one-day-old adults were harvested with Milli-Q water and washed until the supernatant was cleared. Excess water was removed by centrifugation and worm pellets (0.86 to 1.35 grams) were flash frozen and stored at −80°C until analysis. Total fatty acids were quantitatively measured by Creative Proteomics (Shirley, NY) using gas chromatography (GC) with flame ionization detection as follows: to extract fatty acids, worm samples were weighed into a screw-cap glass vial containing tritricosanoin as an internal standard (tri-C23:0 TG) (NuCheck Prep, Elysian, MN). Samples were homogenized and extracted with a modified Folch extraction. A portion of the organic layer was transferred to a screw-cap glass vial and dried in a speed vac. After samples were dried, BTM (methanol containing 14% boron trifluoride, toluene, methanol; 35:30:35 v/v/v) (Sigma-Aldrich, St. Louis, MO) was added. The vial was vortexed briefly and heated in a hot bath at 100°C for 45 min. Following cooling, hexane (EMD Chemicals, USA) and HPLC grade water were added, tubes were recapped, vortexed, and centrifuged to help in the separation of layers. An aliquot of the hexane layer was transferred to a GC vial. GC was performed using a GC-2010 Gas Chromatograph (Shimadzu Corporation, Columbia, MD) equipped with a SP-2560, 100 m fused silica capillary column (0.25 mm internal diameter, 0.2 µm film thickness; Supelco, Bellefonte, PA). Fatty acids were identified by comparison with a standard mixture of fatty acids (GLC OQ-A, NuCheck Prep), which was also used to determine the individual fatty acid calibration curves. Fatty acid composition was expressed as a percent of total identified fatty acids and concentrations as µg/mg of worms.

## RNA interference

Gravid adults were treated with hypochlorite to obtain embryos using standard methods (*Stiernagle, 2006*). Embryos were placed on NGM plates supplemented with 1 mM IPTG and 50 µg/ml carbenicillin seeded with a 10x concentrated bacterial culture expressing the *k09f5.2* (*vit-1*) RNAi clone obtained from the Ahringer library (*Kamath et al., 2001*). Embryos were allowed to grow until adulthood at which time they were treated again with hypochlorite to obtain embryos. The recovered embryos were grown until day 1 of adulthood under the same conditions and collected for ORO staining.

## Lifespan assays

For control animals, ten L4 larvae ($P_0$ generation) grown in a mixed population cultured on 60 mm NGM plates at 20°C were placed onto each of 3–4 60 mm NGM plates (30–40 worms per replicate) seeded with OP50 and supplemented with 50 µM of 5-fluoro-2'-deoxyuridine (FUDR; Sigma Aldrich) to prevent reproduction. Worm survival was assessed every two days and was deemed dead if no movement was detected after gentle prodding with a worm pick. Animals that crawled to the side of the assay plate and died due to desiccation were censored from the experiment. To obtain $F_1$ animals, 2–3 $P_0$ L4 larva were placed onto one seeded 60 mm NGM plate lacking FUDR and grown at 20°C. Next day, following $P_0$ egg playing, the parents were removed and their $F_1$ progeny allowed to grow at 20°C until they reached the L4 larval stage. Ten L4 F1 larvae were placed onto each of 3–4 OP50-seeded 60 mm NGM plates supplemented with FUDR. $F_1$ worm lifespan was assessed in the same manner as the parental generation. Lifespan assays for RNAi treated animals were done in the same manner except that assay plates were supplemented with 50 µM FUDR, 1 mM IPTG and 50 µg/ml carbenicillin and seeded with a 10x concentrated bacterial culture expressing the *k09f5.2* (*vit-1*) RNAi clone or an empty RNAi vector.

## Germ cell rows

All worm strains used for germ cell row counting have the integrated transgene *gld-1::gfp::3xFLAG* from the BS1080 strain in their genetic background. Worms with L3 vulva morphology were identified as described by using Nomarski DIC microscopy at 630x (*Seydoux et al., 1993*) and DAPI stained using the standard whole worm DAPI staining protocol. The stained worms were imaged using a Leica DM5500B microscope with a Hamamatsu camera controller C10600 ORCA-R2. When performing the germ cell row counts, the start of the transition was identified when at least two cells in a row exhibited the crescent-shaped nuclei morphology (*Shakes et al., 2009*).

## Acknowledgements

We are grateful for the generous gift of dafachronic acid from Frank Schroeder and Pooja Gubibanda for advice on its use. We thank Leszek Kotula and Angelina Regua for access to their microscope, Eleanor Maine for thoughtful comments on this manuscript, and Jason Fridley for assistance with statistics. Strains AA1052 and BS1080 were kindly provided by Adam Antebi and Tim Schedl, respectively. We thank the CGC, which is funded by the NIH Office of Research Infrastructure Programs (P40 OD010440), for providing strains. This work was partially supported by a NIHR01GM129135 grant to SEH.

## Additional information

### Funding

| Funder | Grant reference number | Author |
| --- | --- | --- |
| National Institutes of Health | R01GM129135 | Sarah E Hall |

The funders had no role in study design, data collection and interpretation, or the decision to submit the work for publication.

### Author contributions

Maria C Ow, Conceptualization, Formal analysis, Investigation, Writing - original draft; Alexandra M Nichitean, Investigation, Writing - review and editing; Sarah E Hall, Conceptualization, Formal analysis, Funding acquisition, Writing - review and editing

### Author ORCIDs

Sarah E Hall  https://orcid.org/0000-0002-8536-4000

### Decision letter and Author response

Decision letter https://doi.org/10.7554/eLife.61459.sa1
Author response https://doi.org/10.7554/eLife.61459.sa2

## Additional files

### Supplementary files

• Supplementary file 1. Gene expression changes of selected genes in N2 PD$_{Stv}$ compared to N2 CON adults (*Ow et al., 2018* PLoS Genet 14: e1007219).

• Supplementary file 2. Strains used in this study.

• Supplementary file 3. Primer sequences used in this study.

• Transparent reporting form

### Data availability

All data generated or analyzed during this study are included in source files associated with relevant figures.

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

## Appendix 1

### Additional results and discussion

GLP-4 is required globally to affect reproductive plasticity resulting from early life starvation

We have previously shown that postdauer adults exhibit gene expression and reproductive phenotypes that reflect their environmental and developmental history. In adults that experienced starvation-induced dauer formation (PD$_{Stv}$), we showed that gene expression changes of somatically-expressed seesaw genes and their characteristic reduced brood size relative to control adults (CON) required a functional *glp-4* gene (**Ow et al., 2018**). The *glp-4* gene encodes a valyl aminoacyl tRNA synthetase that is expressed in the intestine, somatic gonad, and the germ line (**Rastogi et al., 2015**). The temperature-sensitive *bn2* allele is a partial loss-of-function lesion that likely results in decreased translation in both somatic and germline tissues (**Beanan and Strome, 1992**; **Rastogi et al., 2015**). *glp-4(bn2)* mutants exhibit increased adult lifespan and stress resistance that require crosstalk between the germ line and the soma via endocrine signaling pathways (**Arantes-Oliveira et al., 2002**).

To address whether the expression of *glp-4* in the intestine, somatic gonad, or the germ line affect reproductive plasticity between control and PD$_{Stv}$ adults, we constructed single copy rescuing transgenes with tissue specific promoters using Mos1-mediated single copy insertion (MosSCI). We performed brood size assays of control and PD$_{Stv}$ adults at 15°C, the permissive temperature at which *glp-4(bn2)* is fertile. While the brood size of PD$_{Stv}$ wild-type N2 adults was decreased to near significance ($p = 0.0525$) at the low temperature, the brood size of *glp-4(bn2)* PD$_{Stv}$ adults increased significantly, consistent with our previous observation (**Appendix 1—figure 1**; **Ow et al., 2018**). Expression of *glp-4* under its endogenous promoter partially rescued the increased brood size of PD$_{Stv}$ *glp-4* adults (**Appendix 1—figure 1**). However, limited expression of *glp-4* in the intestine, somatic gonad, or the germ line resulted in a brood size phenotype similar to that of *glp-4(bn2)* mutants (**Appendix 1—figure 1**). These results suggest that the contribution of GLP-4 to the reproductive plasticity of PD$_{Stv}$ adults is not due to its function in a singular tissue (germ line, somatic gonad, or intestine) but rather from multiple locations to promote inter-tissue crosstalk.

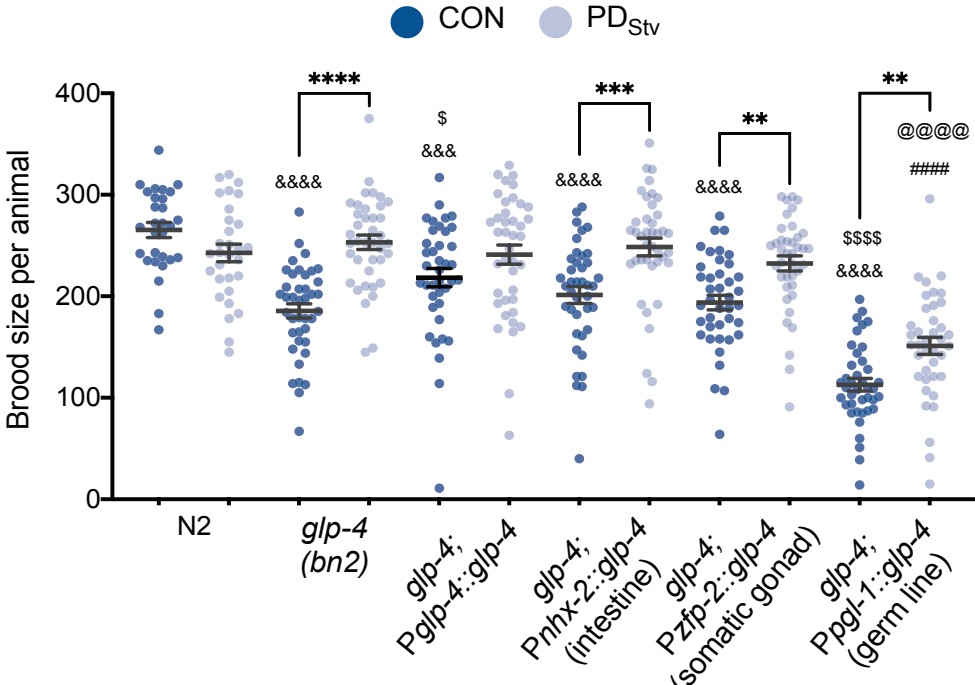

**Appendix 1—figure 1.** GLP-4 is required in multiple tissues to regulate adult reproductive plasticity. Brood size comparison of PD$_{Stv}$ relative to CON in wild-type N2, *glp-4(bn2)*, and tissue-specific MosSCI rescue strains of *glp-4*. Assays were done at 15°C. ** p < 0.01, *** p < 0.001, and **** p < 0.0001 compare PD$_{Stv}$ to CON within a genotype; [&&&] p < 0.001 and [&&&&] p < 0.0001 compare CON of N2, mutant and *glp-4* tissue-specific rescues; [####]p < 0.0001 compares PD$_{Stv}$ of N2 to mutant and rescue strains; [$]p < 0.05 and [$$$$]p < 0.0001 compare CON of *glp-4* and rescue strains; [@@@@]p < 0.0001 compares PD$_{Stv}$ of *glp-4* and rescue strains; one-way ANOVA with Sidak's multiple comparison test. Error bars represent S.E.M. Additional brood size data are provided in *Appendix 1—figure 1—source data 1*.

The online version of this article includes the following source data is available for figure 1:

**Appendix 1—figure 1—source data 1.** GLP-4 is required in multiple tissues to regulate adult reproductive plasticity.

## The set of genes with altered expression in PDstv adults is significantly enriched for DAF-16 Class I and Class II targets

We observed that 18.82% (313 genes; p < 2.52e-91, hypergeometric test) of the reported 1,663 DAF-16 Class I target genes and 13.21% (229 genes; p < 1.96e-37) of the 1,733 PQM-1 Class II target genes significantly overlapped with the 1121 somatically enriched genes that we previously reported to be upregulated in PD$_{Stv}$ adults. We also observed significant overlaps between our previously reported 551 downregulated genes in PD$_{Stv}$ adults with DAF-16 Class I genes (0.54%; nine genes; p < 1.03e-11, hypergeometric test) and PQM-1 Class II genes (4.79%; 83 genes; p < 2.83e-07) (*Tepper et al., 2013*; *Ow et al., 2018*; *Figure 2—figure supplement 1*).

A subset of DAF-16 targets are genes with functions in fat metabolism. DAF-16, along with TCER-1, alter the expression of lipid biosynthesis, storage, and hydrolysis genes to promote adult longevity in animals lacking a germ line (*Amrit et al., 2016*). We also found a significant enrichment DAF-16 and TCER-1 targeted genes in our set of genes with altered mRNA levels in PD$_{Stv}$ adults. Specifically, 47% (7 out of 15) of DAF-16 and TCER-1 target genes predicted to regulate lipid synthesis and storage are upregulated in PD$_{Stv}$ adults (p < 6.86e-06, hypergeometric test). Similarly, 50% (12 out of 24) of lipid hydrolysis genes targeted by DAF-16 and TCER-1 are up-regulated in PD$_{Stv}$ adults (p < 1.16e-09) (*Amrit et al., 2016*; *Ow et al., 2018*; *Figure 2—figure supplement 1*). Additionally, 27% (4 out of 15) of lipid catabolism genes (p < 0.008) and 12% (9 out of 74) of lipid anabolism genes (p < 0.021) that were not identified as targets of DAF-16 or TCER-1 were upregulated in PD$_{Stv}$ adults (*Amrit et al., 2016*; *Ow et al., 2018*). Notably, none of the genes down-regulated in PD$_{Stv}$ adults were represented by the DAF-16 and TCER-1 lipid metabolic target genes (*Amrit et al., 2016*; *Ow et al., 2018*; *Figure 2—figure supplement 1*). This observation was also true for lipid anabolism and catabolism genes that are not targets of DAF-16 or TCER-1.

## Increased dafachronic acid is not sufficient to decrease brood size in postdauer animals

We asked whether the exogenous addition of dafachronic acid (DA) was sufficient to reduce the brood size of PD$_{Stv}$ adults. To test this hypothesis, we induced larvae into dauer by starvation with or without exogenously added 40 nM of $\Delta^7$-DA and measured their brood size. The addition of $\Delta^7$-DA did not affect the fertility of wild-type and *daf-36(k114)* (*Figure 1—figure supplement 1*). However, *daf-9(rh50)* PD$_{Stv}$ adults exhibited a significant decrease in brood size in the presence of exogenous $\Delta^7$-DA (*Figure 1—figure supplement 1A*). DAF-9 is reported to promote a feedback loop of DA production between the neuroendocrine XXX cells and the hypodermis; thus, *daf-9* mutants may be more sensitive to small changes in DA concentration (*Antebi, 2014*). To examine the interaction of the steroid signaling and reproductive longevity pathways, we also examined whether exogenous $\Delta^7$-DA affected the brood sizes of *kri-1(ok1251)* and *tcer-1(tm1425)* PD$_{Stv}$ adults. We found that the brood sizes of *kri-1* and *tcer-1* animals were not significantly affected in this experiment (*Figure 1—figure supplement 1A*). Taken together, these results suggest that increased DA-dependent DAF-

12 signaling is necessary, but perhaps not sufficient, for the reduced brood size phenotype of wild-type PD$_{Stv}$ adults.

## Control adult physiology is dependent upon cultivation conditions

We noted in the aging, brood size, and ORO staining experiments described in the results that F$_1$ progeny of controls exhibited significant differences in assay measurements compared to the parental control population (*Figure 5C*, *Figure 5—figure supplement 3*, *Figure 6*). One possible explanation is that cultivation conditions of the P$_0$ and the F$_1$ and F$_2$ populations differed: P$_0$ animals were obtained from a mixed population consisting of all life stages (embryos to adults), while the composition of the F$_1$ and F$_2$ populations were more synchronous. Previous work has reported that population density affects the progression of worm development due to the type of chemical signals, such as ascarosides, extruded into the environment (*Ludewig et al., 2017*). Pheromones are secreted throughout worm development; thus, we hypothesize a pheromone-dependent mechanism regulating worm physiology based on cultivation conditions may account for the differences in control population. We should emphasize that a statistically significant lower postdauer fecundity is observed regardless of whether the P$_0$ animals were chosen from a mixed population culture or from a homogenous staged population (*Figure 1B*; *Figure 5—figure supplement 2*). Investigating the mechanisms of these differences will be the subject of future work.

## DAF-12 likely acts downstream of TCER-1 and KRI-1 to regulate reproductive plasticity

To further examine the genetic interactions of DAF-12 and TCER-1, we performed epistasis analysis by measuring the brood sizes of control and PD$_{Stv}$ adults in *kri-1(ok1251); daf-12(rh284)* and *tcer-1 (tm1425); daf-12(rh284)* double mutants, and in *kri-1(ok1251); daf-12(rh285)* and *tcer-1(tm1425); daf-12(rh285)* double mutants. For all four double mutants, we continued to observe a significantly increased brood size in PD$_{Stv}$ adults compared to controls, similar to the *daf-12(rh284)* and *daf-12 (rh285)* single mutants (*Figure 1B*; *Figure 1—figure supplement 1B*). The combination of mutations between *kri-1(ok1251)* and *tcer-1(tm1425)* with the *daf-12(rh284)* allele synergistically exacerbated the fertility phenotype of PD$_{Stv}$ adults to a more than 600-fold average compared to each of the single mutants, suggesting that these pathways act in parallel (*Figure 1B*; *Figure 1—figure supplement 1*). However, the brood sizes of the *kri-1(ok1251); daf-12(rh285)* and *tcer-1(tm1425); daf-12 (rh285)* double mutants were statistically indistinguishable from the *daf-12(rh285)* single mutant, instead suggesting that DAF-12 acts downstream of TCER-1 and KRI-1 in the same pathway (*Figure 1B*; *Figure 1—figure supplement 1*). Since the *daf-12* and *tcer-1* mutant strains used in our experiment are not null alleles, we must interpret these epistasis results cautiously. However, we can make some conclusions with respect to the phenotypes of *daf-12(rh284)* and *daf-12(rh285)*, which have previously been characterized for developmental timing and dauer formation defects (*Antebi et al., 1998*; *Antebi et al., 2000*). The *daf-12(rh284)* mutant (P746S lesion in helix 12 of the LBD) displays delayed gonadal development while the *daf-12(rh285)* mutant (Q707stop mutation in the LBD after helix 9) has penetrant heterochronic phenotypes that include delayed gonadal and extragonadal developmental events (*Antebi et al., 1998*; *Antebi et al., 2000*). In addition, our brood assay results show that the *rh285* allele has a more severe phenotype than *rh284* in terms of reproduction (*Figure 1B*). These phenotypes are perhaps due to the differences in the nature of LBD disruption that have not been characterized.

We have also taken a second approach to dissect the roles of steroid signaling and the reproductive longevity pathway in the regulation of PD$_{Stv}$ fecundity. First, we showed that *tcer-1(tm1425)* PD$_{Stv}$ animals do not exhibit the delay in germline proliferation that we observed in wild-type PD$_{Stv}$ animals (*Figure 7*; *Figure 7—figure supplement 1*). We also observed that there is no difference in total germ cell rows, mitotic cells rows, and meiotic cell rows between *kri-1(ok1251)* control and PD$_{Stv}$ animals, indicating that KRI-1 is required for the observed delay in germline proliferation in wild-type PD$_{Stv}$ animals (*Appendix 1—figure 2*). Moreover, *kri-1* control animals have fewer total and meiotic cell rows than wild-type controls, suggesting that KRI-1 may also be required for proper germline development and the onset of germline proliferation in a favorable environment (*Appendix 1—figure 2*). Together, our results favor a model in which DAF-12 acts downstream of KRI-1 and

TCER-1 due to the greater severity of the *daf-12(rh285)* allele over the *daf-12(rh284)* allele in the epistasis.

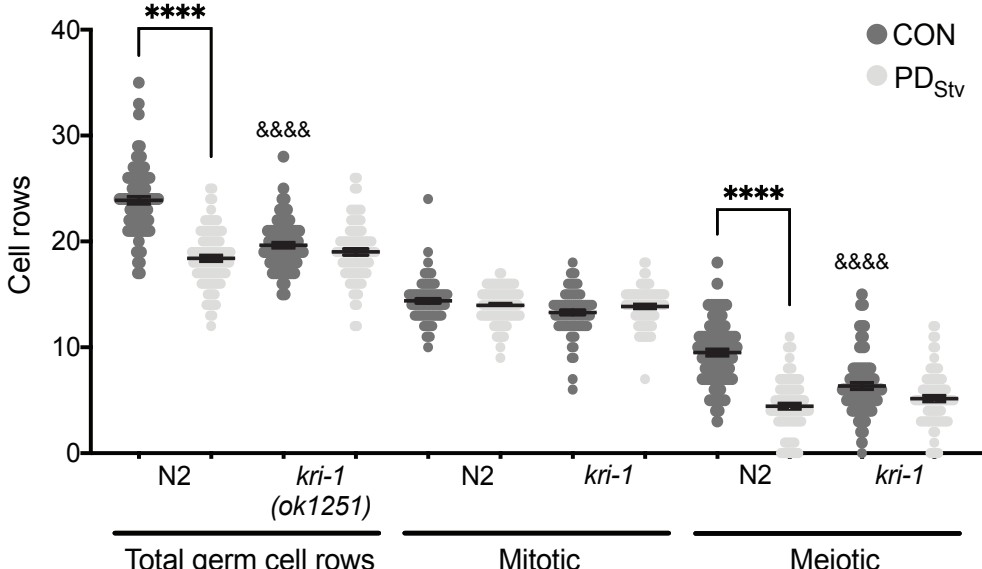

**Appendix 1—figure 2.** KRI-1 regulates the onset of germline proliferation. Total, mitotic, and meiotic cell rows in control and PD_Stv animals exhibiting L3 vulva morphology in wild-type N2 and *kri-1(ok1251)*. \*\*\*\* p < 0.0001 compares of CON and PD_Stv within a genotype; &&&&p < 0.0001 compares N2 CON to mutant CON within a type of cell row count; one-way ANOVA with Sidak's multiple comparison test. Error bars represent S.E.M. Additional data are provided in *Appendix 1—figure 2—source data 1*.

The online version of this article includes the following source data is available for figure 2:

**Appendix 1—figure 2—source data 1.** Total, mitotic, and meiotic cell rows in control and PDStv animals in wild type and kri-1(ok1254).

## PD_Stv hermaphrodites do not have an oogenesis or reproductive defect

To test whether the decreased fecundity of PD_Stv adults was the due to a reduction in sperm number as a consequence of the dauer experience as we previously reported (*Ow et al., 2018*), we compared the brood sizes of wild-type self-fertilized CON and PD_Stv hermaphrodites to those mated with control wild-type males. We found that the sperm supplied from control males resulted in a statistically insignificant difference between the brood sizes of mated CON and PD_Stv hermaphrodites, indicating that a contributing factor in the decreased fecundity in PD_Stv animals is a limitation in sperm number and not an oogenesis or reproduction defect (*Appendix 1—figure 3*).

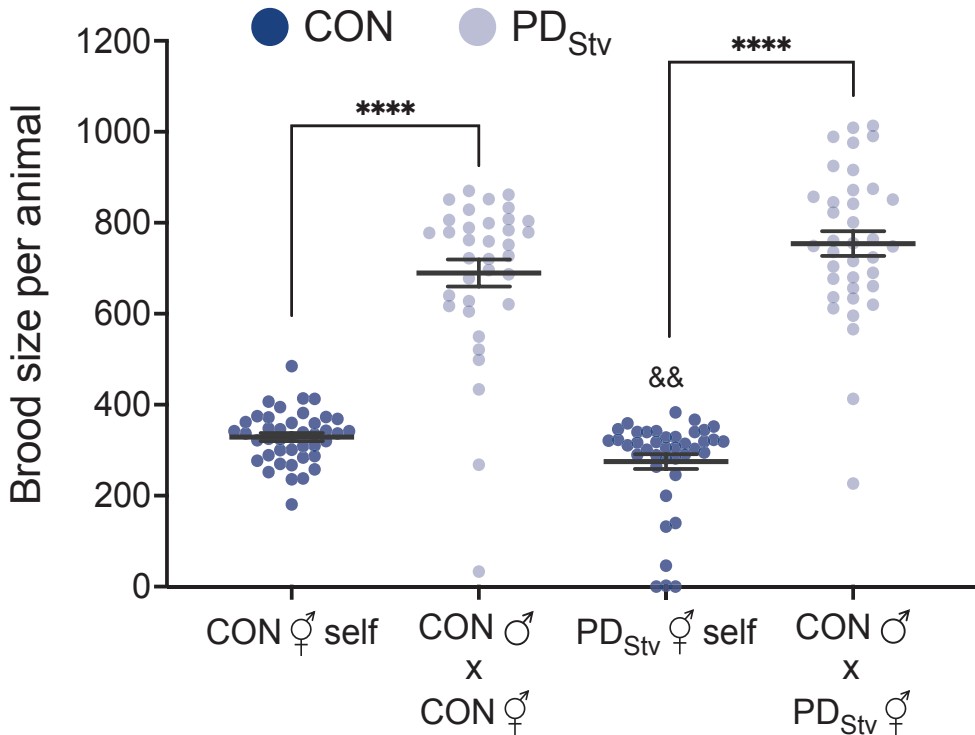

**Appendix 1—figure 3.** Decreased fecundity in postdauer animals results from a reduction of sperm available for self-fertilization. Brood size of wild-type N2 CON and PD$_{Stv}$ self-fertilized hermaphrodites (⚥) and CON or PD$_{Stv}$ hermaphrodites mated with CON males (♂). **** p < 0.0001 compares self-fertilized CON or PD$_{Stv}$ hermaphrodites to CON or PD$_{Stv}$ hermaphrodites mated to CON males; &&p < 0.01 compares self-fertilized CON to self-fertilized PD$_{Stv}$ hermaphrodites; Student's *t*-test. Error bars represent S.E.M. Additional data are provided in *Appendix 1—figure 3—source data 1*.

The online version of this article includes the following source data is available for figure 3:

**Appendix 1—figure 3—source data 1.** Decreased fecundity in postdauer animals results from a reduction of sperm available for self-fertilization.

## Additional methods

### Plasmid construction and transgenic strains

To clone P*glp-4::glp-4 cDNA::gfp::glp-4* 3'UTR, primers sets MO2454 and MO2455; MO2516 and MO2507; MO2508 and MO2468; MO2469 and MO2470; and MO2471 and MO2472 were used to amplify pUC19, 1.2 kb of the *glp-4* promoter, the *glp-4* cDNA (using a cDNA library prepared from total RNA of a mixed N2 population as template), the *gfp* gene (using Fire vector pPD95.75 as the template), and 500 bp of the *glp-4* 3'UTR, respectively, using Phusion DNA polymerase (NEB). Assembly of PCR products was done using the NEBuilder HiFi DNA Assembly (E2621) to generate pMO555. P*glp-4::glp-4 cDNA::gfp::glp-4* 3'UTR was cloned into the pCFJ151 MosSCI vector to generate plasmid pMO557 using primer sets MO2499 and MO2500 and MO2514 and MO2504 to amplify pCFJ151and P*glp-4::glp-4 cDNA::gfp::glp-4* 3'UTR (using pMO555 as template), respectively, using Phusion DNA polymerase (NEB) and assembled with the NEBuilder HiFi DNA Assembly. To clone P*nhx-2::glp-4 cDNA::gfp::glp-4* 3'UTR, overlap extension PCR was done with primers MO2544 (*Xma*I) and MO2533 (with genomic DNA as template); MO2534 and MO2532 (*Sbf*I) (with pMO555 as template). The resulting amplicon was digested with *Xma*I and *Sbf*I and cloned into pUC19 to generate pMO562. P*nhx-2::glp-4 cDNA::gfp::glp-4* 3'UTR was cloned into pCFJ151 using the NEBuilder HiFi DNA Assembly and PCR products from primers MO2509 and MO2504 (with pMO562 as template) and primers MO2499 and MO2500 (with pCFJ151 as template) to create

pMO565. To clone P*pgl-1::glp-4 cDNA::gfp::glp-4* 3'UTR, overlap extension PCR was done with primers MO2545 (*Xma*I) and MO2536 (with genomic DNA as template); MO2537 and MO2532 (*Sbf*I) (with pMO555 as template). The resulting amplicon was digested with *Xma*I and *Sbf*I and cloned into pUC19 to generate pMO559. P*pgl-1::glp-4 cDNA::gfp::glp-4* 3'UTR was cloned into pCFJ151 using the NEBuilder HiFi DNA Assembly and PCR products from primers MO2510 and MO2504 (with pMO559 as template) and primers MO2499 and MO2500 (with pCFJ151 as template) to create pMO566. To clone P*zfp-2::glp-4 cDNA::gfp::glp-4* 3'UTR, overlap extension PCR was done with primers MO2546 (*Xma*I) and MO2538 (with genomic DNA as template); MO2539 and MO2532 (*Sbf*I) (with pMO555 as template). The resulting amplicon was digested with *Xma*I and *Sbf*I and cloned into pUC19 to generate pMO560. P*zfp-2::glp-4 cDNA::gfp::glp-4* 3'UTR was cloned into pCFJ151 using the NEBuilder HiFi DNA Assembly and PCR products from primers MO2511 and MO2504 (with pMO560 as template) and primers MO2499 and MO2500 (with pCFJ151 as template) to create pMO567. Plasmids pMO557, pMO565, pMO566, and pMO567 were used to generate single copy insertions, *pdrSi1* (P*glp-4::glp-4 cDNA::gfp::glp-4 3'UTR*), *pdrSi2* (P*nhx-2::glp-4 cDNA::gfp::glp-4 3'UTR*), *pdrSi3* (P*zfp-2: :glp-4 cDNA::gfp::glp-4 3'UTR*), and *pdrSi4* (P*pgl-1::glp-4 cDNA::gfp::glp-4 3'UTR*) by Mos1-mediated single copy insertion (MosSCI) (*Frøkjaer-Jensen et al., 2008*). Following integration into *unc-119(ed3)* animals segregated from EG6699 [*ttTi5605 II; unc-119(ed3) III; oxEx1578 (eft-3p::gfp + Cbr-unc-119)*] that have lost the *oxEx1578* array, MosSCI insertions were genetically crossed into the *glp-4(bn2)* background. Primer sequences used in plasmid construction are listed in *Supplementary file 3*.

## Dafachronic acid supplementation

The $\Delta^7$ form of dafachronic acid ($\Delta^7$-DA) (a kind gift from Frank Schroeder) was added to a freshly grown culture of *E. coli* OP50 at a concentration of 40 nM and immediately seeded onto NGM plates. NGM plates supplemented with an equivalent volume of ethanol (the $\Delta^7$-DA solvent) to those of the $\Delta^7$-DA-supplemented NGM plates were used as the control plates. Seeded plates were allowed to dry overnight before use.

## DAF-16 localization

DAF-16::GFP localization in CON and PD$_{Stv}$ one-day-old adults was examined using strains CF1139 (*daf-16(mu86) I; muIs61 [(pKL78) daf16::gfp + rol-6(su1006)]*) and TJ356 (*zIs356 [daf-16p::daf-16a/b:: gfp + rol-6(su1006)]*) (*Henderson and Johnson, 2001*; *Lin et al., 2001*). Worms were imaged using a Leica DM5500B microscope with a Hamamatsu camera controller C10600 ORCA-R2.

## RNA extraction

Total RNA was extracted using TRIzol Reagent (Life Technologies). Four volumes of TRIzol reagent were added to a frozen worm pellet followed by vigorous vortexing for 20 min. Samples were centrifuged in a tabletop centrifuge at maximum speed and the cleared supernatant was transferred to a fresh tube. The RNA was precipitated with equal volume of isopropanol at −80°C for at least 30 min. RNA pellets were washed with cold 75% ethanol, dried, and resuspended in RNase-free water.

## Quantitative reverse transcription real-time PCR

Total RNA was treated with DNaseI (NEB) and processed with Superscript IV First Strand Synthesis Systems (Life Technologies) using oligo (dT) primers following the recommendations of the manufacturer. Real-time PCR was done with iTaq Universal SYBR Green Supermix (BioRad) according to the recommendations of the manufacturer. C$_t$ normalization was done using *act-1*. All primer sequences are listed in *Supplementary file 3*.

## RNA interference

Gravid adults were treated with sodium hypochlorite to obtain embryos using standard methods (*Stiernagle, 2006*). Embryos were placed on NGM plates supplemented with 1 mM IPTG and 50

µg/ml carbenicillin seeded with a 10x concentrated bacterial culture expressing the *k09f5.2* (*vit-1*) RNAi clone obtained from the Ahringer library (*Kamath et al., 2001*) or an empty RNAi vector. Embryos were allowed to grow until adulthood at which time they were treated again with sodium hypochlorite to obtain embryos. For obtain postdauer animals, worms were grown at 20°C for 2 weeks to induce starvation-induced dauer formation. Dauers were selected using 1% SDS treatment (*Stiernagle, 2006*). The recovered embryos or dauers were grown on the appropriate RNAi plates until the L4 stage or day 1 of adulthood and used for brood assays, lifespan assays or ORO staining.

## Transgenerational brood assays

For control animals, ten to fifteen L4 larvae ($P_0$ generation) grown in a mixed population propagating on 60 mm NGM plates at 20°C were placed singly onto 35 mm or 60 mm NGM plates. $P_0$ parents were transferred daily to fresh plates until egg laying ceased. Ten to fifteen L4 $F_1$ progeny from days 1 or 2 of the $P_0$ parents' egg laying period, were picked from each brood assay plate and placed individually onto 35 mm or 60 mm NGM plates. $F_1$ parents were transferred daily until egg laying ceased. Ten to fifteen L4 $F_2$ progeny from day 1 or 2 of the $F_1$ parents' egg laying period, were singled from each brood assay plate and placed onto individual 35 mm or 60 mm NGM plates. $F_2$ parents were transferred daily until egg laying ceased. To obtain postdauer animals, a well-fed 60 mm worm plate was allowed to starve for 1–2 weeks at 20°C. Starved worms were washed with M9 buffer. Dauers were selected with 1% SDS treatment (*Stiernagle, 2006*) and allowed to recover for 1 day at 20°C on 60 mm NGM plates. Ten to fifteen postdauer L4 larvae ($P_0$ generation) were singled onto 35 mm or 60 mm NGM plates in the same manner as control animals. The brood sizes of $F_1$ and $F_2$ postdauer generations were obtained in the same way as $F_1$ and $F_2$ control animals. All progenies were counted 4 days following parental transfer. Eleven biologically independent assays were performed.

## Mating assays

Three N2 young adult males were placed onto each of ten 35 mm NGM plates seeded with a drop of *E. coli* OP50 containing one N2 hermaphrodite L4 larvae per plate and allowed to mate for 24 hr at 20°C. Males were removed and hermaphrodites transferred daily to 35 mm seeded NGM plates until egg laying ceased. Animals that did not produce any male progeny due to the failure of a successful mating event were censored from the experiment. Live progenies were counted four days following the transfer of their parent to fresh NGM plates. Mating assays were conducted in parallel with self-mated animals. Four biologically independent assays were performed.

## *kri-1* germ cell rows

Wild-type and *kri-1(ok1251)* worms with L3 vulva morphology were identified as described by using Nomarski DIC microscopy at 630x (*Seydoux et al., 1993*) and DAPI stained using the standard whole worm DAPI staining protocol. The stained worms were imaged using a Leica DM5500B microscope with a Hamamatsu camera controller C10600 ORCA-R2. When performing the germ cell row counts, the start of the transition was identified when at least two cells in a row exhibited the crescent-shaped nuclei morphology (*Shakes et al., 2009*).

