## [Decision Letter]

**Acceptance summary:**

Overall, we believe that this current study makes an important contribution and builds on your previous work on the mechanisms of how early experience impacts later development and the transmission of that "memory" to progeny. We recognize that is an exciting and emerging area of research and that this current study provides key insights into how early starvation-induced memory affects fat storage and fecundity in adults, reveals the steroid signaling pathways and molecules involved, and how the starvation memory is transmitted to progeny in a germline nuclear RNAi-dependent manner. The reviewers and I agree that the experiments were well executed and the conclusions well supported.

**Decision letter after peer review:**

Thank you for submitting your article "Somatic aging pathways regulate reproductive plasticity in *Caenorhabditis elegans*" for consideration by *eLife*. Your article has been reviewed by 3 peer reviewers, and the evaluation has been overseen by John Kim as the Reviewing Editor and Kathryn Cheah as the Senior Editor. The following individual involved in review of your submission has agreed to reveal their identity: Victor Ambros (Reviewer #2).

The reviewers have discussed the reviews with one another and the Reviewing Editor has drafted this decision to help you prepare a revised submission.

Summary:

All three reviewers and the editors think that, in principle, this manuscript is suitable for publication in *eLife*. It nicely builds on your previous work to provide novel insights into the role of steroid hormone signaling in the differences in fecundity and lipid storage for animals that experience early life stresses compared to those that do not. In addition, the study addresses how such signals may be transmitted to the next generation, resulting in phenotypes such as increased fat storage. However, there was also strong consensus that several critical revisions must be carried out before it is acceptable for publication.

Essential revisions:

After consultation, the three reviewers and I agree that the three critical points raised by Reviewer #2 must be addressed. In addition, reviewer #1 and #3 both had significant issues with the MUFA/OA experiments in terms of both presentation and interpretation and agreed that two fairly simple experiments would help clarify the conclusions of the results. While we hope that you will consider all of their comments, I have summarized below the 4 major revisions that we would like you to address:

1. The interpretation of the fecundity data: Reviewer #2 poses an alternative, and possibly more parsimonious, interpretation of the fecundity data. If you agree, please consider rewriting this section, or at least incorporating this plausible alternative interpretation of the fecundity data (and better presenting the data so that they don't exaggerate or mislead the effects due to small numbers as noted by Reviewer #3).

2. We ask that you consider some simple genetic analysis to test whether the fecundity defects are due to limiting sperm as you hypothesize. These experiments would directly provide empirical support for this hypothesis.

3. The F1 phenotypes such as increased lipid storage are very interesting and suggests that such metabolic reprogramming may confer the F1s a physiological advantage. Therefore, it would be informative to show some physiological consequence of the inherited F1 phenotype such as its impact on fecundity and longevity.

4. Reviewers #1 and #3 found the MUFA/OA experiments quite confusing and contradictory. Please address their concerns with text revisions plus some simple experiments that Reviewer #1 suggests – i.e. measuring MUFAs/OA in control and PDStv adults and seeing if exogenous OA addition rescues the reduced fecundity phenotypes of the fat-5/6/7 double mutants.

*Reviewer #1:*

The authors have published multiple papers on the topic of post-dauer phenotypic plasticity in *C. elegans*. They previously showed that post-dauer adults have distinct gene expression profiles that are dependent upon the nature of the dauer-inducing stressor and correlated with opposing brood size phenotypes. Here they report studies designed to reveal the molecular basis for the reduced fecundity phenotype observed in wild-type adults that traversed dauer due to starvation during early larval development (referred to as PDStv). They show that many of the same molecules and pathways that mediate germline-ablation-induced longevity (DA/DAF-12 signaling, TCER-1, and lipid metabolism genes) are also required for reduced fecundity in PDStv adults. Oil Red O staining reveals that PDStv adults have reduced intestinal fat stores concomitant with increased embryo fat staining. This reapportioning of fat stores requires DAF-12 and vitellogenins. Delayed gametogenesis, which is associated with the reduced fecundity of PDStv adults, is also mediated by DA and TCER-1. Finally, they show that PDStv adults give rise to F1 progeny with increased intestinal fat stores compared to F1 progeny of control animals that did not traverse dauer, and this transgenerational phenotype requires HRDE-1.

Overall the experiments are well-executed, and the data are convincing. In addition, the authors have done well to identify several genes that participate in post-dauer reproductive plasticity. However, in my opinion the study does not yield much mechanistic insight into this process. Moreover, there is one aspect of the data that seems contradictory to me and needs to be either resolved experimentally or explained more clearly.

1. Figure 1: since DHS-16 also participates in DA biosynthesis (Wollam et al. PLOS Biology 2012), a dhs-16 mutant should also be tested here. Since DHS-16 is required for delta-7-DA biosynthesis but not delta-1,7-DA biosynthesis, this result could help pinpoint the specific DA involved in this process.

2. Lines 265-286: I may be missing something, but I am confused about the role of MUFAs/OA in the reduced fecundity phenotype of PDStv adults. In lines 265-266, it is stated that "…MUFAs are required for the decreased fertility phenotype…" This is consistent with the observation that double-mutant combinations of fat-5/6/7 alleles suppress the phenotype (Figure 3D). However, based on the OA supplementation experiment (Figure 3E), the authors conclude in lines 280-281 that "…OA is a limiting factor for reproduction after passage through the starvation-induced dauer stage…" Is there an optimal concentration of MUFAs and/or OA that inhibits fecundity? Measuring MUFAs/OA in control and PDStv adults would help to resolve this question.

3. Related to the above point, does exogenous OA supplementation rescue the reduced fecundity phenotype in fat-5/6/7 double mutants?

4. Does vit-1 RNAi affect the brood size of PDStv adults?

*Reviewer #2:*

Ow et all report findings that build upon their previous work (Ow et al. 2018), where they had shown that *C. elegans* adults that had developed through the alternative L3 larval stage, the dauerlarva, displayed altered fecundity and altered gene expression programs compared to control animals that had not undergone the dauer larva stage (ie had developed continuously from egg to adult). Moreover, in that previous paper, the authors reported that the phenotype of postdauer adults, including the aspects of their transcriptional program, depended on the particular stress used to induce dauer larva entry – high pheromone vs starvation – during the pre-dauer larval development. This current study builds upon those previous findings to explore the mechanisms whereby starvation-induced dauer larva arrest, followed by post-dauer L3 and L4 development, can affect the physiology of the worm and its progeny.

The authors report here that steroid hormone signaling is involved in the differences in fecundity and fat storage in postdauer adults compared to control (non-dauer) adults. They also report that the effects of starvation-induced dauer larva development can be transmitted to the next generation to affect fat storage in the progeny. The authors emphasize that these effects of life history on reproductive capacity and lipid metabolism are associated with the same steroid signaling pathways that affect longevity in response to reproductive stress, and suggest that, ".…ancestral starvation memory is inherited and may be the result of crosstalk between somatic and reproductive tissues mediated by the germline nuclear RNAi pathway." These findings are interesting, particularly the epigenetic programming of apparently adaptive metabolic physiology in progeny, dependent on the life history of the parent.

The findings of the paper, broadly speaking, are reasonably well supported by the data, in that life history dependent alterations in fecundity and lipid metabolism are well documented, and the transmission of phenotype to the F1 is also well supported by the data. The effects of mutations in steroid hormone signaling and RNAi mutants are also well demonstrated. So, the essential findings of the paper are robust, and their implications are undoubtedly important in terms of advancing understanding of adaptive interplay between development and physiology.

However, there are some significant concerns that should be addressed:

1) Alternative interpretations of certain important findings were not considered.

These concerns have to do with the conclusions summarized in Line 29: ".… steroid hormone signaling promotes fat reallocation in postdauer adults…" and Line-109: "Dafachronic acid-dependent DAF-12 signaling is required for decreased fecundity after starvation-induced dauer formation".

The above conclusions are derived chiefly from the data in Figures 1 and 4. In Figure 1, it is apparent that daf-9(rh50) and daf-36(k114) mutants that had traversed the starvation-induced dauer larva exhibited a significant increase in brood size compared to animals of the same genotypes that had not experienced starvation (or dauer larva arrest). This is an exceedingly interesting finding, since these mutants seem to be behaving opposite to the wild-type in this regard; wild type post dauer animals have a reduced brood size compared to controls. The authors' interpretation of this finding – that steroid hormone signaling causes the reduced fecundity in wild type postdauer adults – is a bit skewed towards one point of view at the expense of an alternative (and arguably more valid) interpretation. The alternative view comes from considering the raw data in the dataset supporting Figure 1 (83372_0_data_set_1656224_q6n6fh.xlsx). In the dataset, it is apparent that the brood sizes of daf-12(lf) daf-9(lf) and daf-36(lf) non-dauer controls are much lower than wild type non-dauer controls. So, it appears that steroid hormone signaling is somehow required for full fecundity in non-dauer adults. Strikingly, the brood size defects of these mutants are essentially suppressed in postdauer animals (see 83372_0_data_set_1656224_q6n6fh.xlsx). Therefore, these results could be interpreted to mean that daf-12, daf-9, and daf-36 are actually more critical for full brood size during continuous development than they are during postdauer development. In other words, one could say that "… daf-12 signaling is required for full fecundity during continuous development, and is relatively dispensable during postdauer development." Similarly, Figure 4 shows that postdauer daf-12 mutants are suppressed for the reduced fat storage that these mutants display when developed continuously (without starvation or dauer arrest). Indeed, fat levels in postdauer daf-12 mutants appear to be restored to essentially the same levels as exhibited by postdauer wild type adults. This finding can be interpreted to suggest that daf-12 is not required for normal fat metabolism in postdauer animals (or at least daf-12 is less critical for fat storage in postdauer adults than in non-dauer adults). Note that this way of framing the conclusions from these data is the opposite to how the authors state their conclusions (that daf-12 regulates fat storage and fecundity of postdauer adults).

2) An instance where an additional experiment could have provided critical tests of otherwise relatively speculative interpretations.

In this paper, following on from initial findings reported in Ow et al. 2018, the authors explore the phenomenon wherein postdauer animals can have altered numbers of germ cells at defined stages of larval development, suggesting differences in the timing of key steps in germline proliferation. The authors suggest that this altered germline proliferation program could affect brood size by altering the number of sperm available for self-fertilization. This is a very interesting hypothesis, that was unfortunately not tested directly by crossing the postdauer hermaphrodites to males to determine if sperm are indeed limiting.

3) Another instance where an additional experiment could have provided critical tests of otherwise relatively speculative interpretations.

Another very interesting observation reported here is that the F1 progeny of postdauer adults, on average have more stored fat than F1 progeny of continuously-developed adults. This is the reverse of the P0 situation, where postdauer adults have less stored fat than controls (Figure 6). This inversion between P0s and F1s is absent in hrde-1 or prg-1 mutations, which are defective in RNAi inheritance. These data support the conclusion that small RNA mediated gene silencing could mediate the reprogramming of progeny physiology. What is missing is a demonstration of whether or not the reprogrammed F1 animals manifest any consequences from the reprogramming. In particular, Does the increased lipid storage in F1 animals confer upon them an altered fecundity and/or longevity?

*Reviewer #3:*

In this manuscript, Ow et al. seek to further elucidate the pathways involved in limiting reproductive output (brood size) after early life stress in the form of starvation-induced dauer in *C. elegans*. Using genetic analysis, they dissected several relevant pathways, including steroid signaling and fat metabolism, that may play a role in limiting brood size in post-dauer adults after starvation (PDst adults). They also examined the accumulation of fat by Oil Red O staining. They found that in the parental generation, fat levels decreased in the intestine of PDst adults. Interestingly, progeny in the F1 generation exhibited an increase in fat levels compared to WT counterparts, and this elevation was reversed by the F2 generation. This intergenerational phenomenon is dependent on germline RNAi factors that have previously been demonstrated to mediate transgenerational inheritance. Overall, the authors tackled an interesting and novel biological question using a system well established and pioneered by their lab. However, the data presented are somewhat disjointed, and at times seemingly contradict each other. Parts of the paper is difficult to follow and the conclusions are unclear.

1. In general, many of the strongest effects on brood size from the genetic analysis (Figures 1-2) are occurring in strains with very small brood sizes to begin with. Thus, plotting the data as log2 fold change normalized between PDst adults and control risks visually exaggerating the effects and misleading the reader. Although the raw data are included and the reader can find out the extreme differences in brood sizes of the mutant strains (for example, daf-12 mutants), the lack of comments / discussion by the authors made the fold change figure misleading. The authors should consider plotting the brood size data (instead of just the fold change), so this difference can be easier to visualize. Although this does not necessarily negate the effects claimed by the authors, the authors need to clearly indicate that some of the strains have greatly reduced brood sizes when cultured under normal condition. Due to this caveat, the genetic interpretation is also unclear. One could argue that the transient dauer arrest somehow reverses the brood size defects of the mutants, rather than that the mutation suppresses the mild reduction on brood size caused by dauer arrest.

2. Similar to comment 1, the authors do not address why PDst should increase the brood size of the mutant strains. It would be logical that if a factor, like DAF-12, was required for the decrease in fertility of PDst adults that in this mutant strain there be no change in brood size after PDst, however in many cases the authors are actually seeing an increase in brood size rather than no change. Is this likely to be an artifact of the low brood size that these strains have under normal condition? It is interesting that the authors previously observed an increase in brood size in post-dauer adults forced into dauer by crowding, but this point is not discussed. Either way, the authors should discuss their interpretation of the brood size increase in PDst adult mutant strains in the text.

3. It would be interesting to investigate if there is any interaction between steroid signaling, TCER-1 and KRI-1 in regulating the low fecundity phenotype. That would help make the manuscript more coherent and easier to follow. In its current state, the earlier findings are not well-connected with the latter findings.

4. DAF-16 plays a critical role in regulating stress-mediated phenotypes. In fig-2B, the authors investigated the brood size of a strain that constitutively localizes DAF-16 in the nucleus. It would be helpful to include the brood size of daf-16(-) worms in PDstv/CONsv to definitely rule out the possibility of DAF-16 involvement.

5. At the beginning of the result section, in lines 111-117, the authors discuss the changes in gene expression that are shared between PDst adults and glp-1 germliness mutants, however they do not show the results of the comparison. Although the RNA-seq experiments were from a previous paper from the lab, the authors should still show the comparison results to convince readers that the similarities between PDst adults and glp-1 warrant investigation. For example, they could show a gene ontology analysis to highlight the pathways that are in common, or show Venn diagrams indicating significant overlap. Without the comparison data, it is difficult for the reader to assess how similar PDst adults and glp-1 pathways really are.

6. The genetic data in Figure 3 was especially difficult to interpret, and the authors need to develop the discussion of this figure more thoroughly. Data shown in Figure 3A suggests that none of the factors (with the exception of sbp-1) show a robust suppression of the PDst fertility phenotype. This is also confusing because the authors include several different alleles of nhr-49 mutants, and see effects in some, but not all of the mutants. Moreover, some mutants are gain-of-function and some loss-of-function, but the authors have not explained why they would expect to see the same phenotype in both GOF and LOF alleles, so overall the results are difficult to interpret. The data become more convincing moving onto Figure 3C. Nevertheless, the authors need a more thorough discussion and consider the inconsistency in their data and interpret the results accordingly.

7. For Figure 3E, the authors state their hypothesis in the text (around line 274) that oleic acid (OA) supplementation should further decrease the brood size after PDst since they just found in Figure 3D that the loss of fat metabolism genes (that produce OA) could rescue the decreased brood size phenotype in PDst adults. This hypothesis is logical from their data, however the authors see the opposite of this result since all strains show an increase in brood size in Figure 3E when on OA. Of course surprising results occur, but the authors do not attempt to reconcile the discrepancy between their hypothesis and the results, leaving the reader very confused. They conclude that OA may be a limiting factor for PDst reproduction, but this is in contrast to their previous conclusion that reducing factors that produce OA with genetic mutants (hence limiting OA) would be beneficial to PDst reproduction. This discussion needs to be clarified. Additionally, the authors do not show the effect of OA on WT worms not exposed to PDst, which could be an important control and perhaps help the authors explain some of their contradictory findings.

8. Following comment 7, it might be helpful to determine whether induction of the fat genes indeed result in changes in MUFAs.

9. The authors show that PDstv adults have increased lipid storage in embryos, even in fat-6(-); fat-7(-) mutants. Do the authors consider increased lipid storage in embryos to be a general phenomenon of low lipid storage mutants?

10. It may be interesting to check the expression of the desaturases in daf-12(-), tcer-1(-) and kri-1(-) worms to tie in the role of these desaturases in regulating low fecundity phenotype. This will also help to connect the earlier results with the fat phenotype.

11. The authors observed that depletion of VIT-1 resulted in significantly increased intestinal fat deposit (Figure 4C and 4D). Since vitellogenins are known to carry fat from the intestine to the developing oocytes, its depletion would be expected to stop the transfer of fat from the intestine to the oocytes, which could result in increased intestinal fat. The authors should consider including this interpretation in the text.

12. The authors suggested that germ cell numbers in daf-12(-) and tcer-1(-) could be connected to their brood sizes in postdauer worms. However, since the difference in germ cell numbers appears to be very small, and the change in brood size appears to be very large. Authors should comment on that.

13. Lastly, the paper at its current state does not read like a coherent paper, but rather it reads like two separate sections.

---

## [Author Response]

Reviewer #1:[…] Overall the experiments are well-executed, and the data are convincing. In addition, the authors have done well to identify several genes that participate in post-dauer reproductive plasticity. However, in my opinion the study does not yield much mechanistic insight into this process. Moreover, there is one aspect of the data that seems contradictory to me and needs to be either resolved experimentally or explained more clearly.1. Figure 1: since DHS-16 also participates in DA biosynthesis (Wollam et al. PLOS Biology 2012), a dhs-16 mutant should also be tested here. Since DHS-16 is required for delta-7-DA biosynthesis but not delta-1,7-DA biosynthesis, this result could help pinpoint the specific DA involved in this process.

We performed brood size experiments using *dhs-16(tm1890)* CON and PD_Stv_ adults as suggested. We observed that the reduced brood size of PD_Stv_ animals is abrogated in the absence of DHS^-^16, suggesting the biosynthesis of delta-7-DA is involved in this process (see Figure 1B).

2. Lines 265-286: I may be missing something, but I am confused about the role of MUFAs/OA in the reduced fecundity phenotype of PDStv adults. In lines 265-266, it is stated that "…MUFAs are required for the decreased fertility phenotype…" This is consistent with the observation that double-mutant combinations of fat-5/6/7 alleles suppress the phenotype (Figure 3D). However, based on the OA supplementation experiment (Figure 3E), the authors conclude in lines 280-281 that "…OA is a limiting factor for reproduction after passage through the starvation-induced dauer stage…" Is there an optimal concentration of MUFAs and/or OA that inhibits fecundity? Measuring MUFAs/OA in control and PDStv adults would help to resolve this question.

Given the upregulation of *fat* genes in PD_Stv_ adults, and their requirement for the decreased PD_Stv_ brood size phenotype, we originally hypothesized that increased OA levels promoted a decrease in progeny. However, our results clearly indicated that supplementing the diet of PD_Stv_ animals with OA increases their brood size, disproving our hypothesis. Our new hypothesis is that OA is limiting for reproduction in PD_Stv_ animals, by acting either as a signaling molecule or as a nutrient. We edited the text to make this distinction clearer. In addition, we performed fatty acid analysis in CON and PD_Stv_ adults. We found that OA levels are unchanged, but levels of ALA and DGLA are significantly different between the two populations (see Figure 4—figure supplement 1C). DGLA has been shown to trigger ferroptosis in the *C. elegans* germline, and OA can abrogate the effects of DGLA (Perez et al. 2020 Dev Cell). We are currently investigating how the relative levels of these fatty acids may influence PD_Stv_ adult brood size.

3. Related to the above point, does exogenous OA supplementation rescue the reduced fecundity phenotype in fat-5/6/7 double mutants?

We performed the suggested experiment, and brood size was only increased in the *fat-5; fat-6* double mutant, suggesting that *fat-7* is required for the OA-dependent increase in brood size (see Figure 4A).

4. Does vit-1 RNAi affect the brood size of PDStv adults?

Decreased vitellogenesis through RNAi knockdown of *vit-1* does not affect the brood size of PD_Stv_ adults. However, the brood size of the CON adults was decreased by *vit-1* RNAi (see Figure 5—figure supplement 2).

Reviewer #2:[…] There are some significant concerns that should be addressed:1) Alternative interpretations of certain important findings were not considered.These concerns have to do with the conclusions summarized in Line 29: ".… steroid hormone signaling promotes fat reallocation in postdauer adults…" and Line-109: "Dafachronic acid-dependent DAF-12 signaling is required for decreased fecundity after starvation-induced dauer formation".The above conclusions are derived chiefly from the data in Figures 1 and 4. In Figure 1, it is apparent that daf-9(rh50) and daf-36(k114) mutants that had traversed the starvation-induced dauer larva exhibited a significant increase in brood size compared to animals of the same genotypes that had not experienced starvation (or dauer larva arrest). This is an exceedingly interesting finding, since these mutants seem to be behaving opposite to the wild-type in this regard; wild type post dauer animals have a reduced brood size compared to controls. The authors' interpretation of this finding – that steroid hormone signaling causes the reduced fecundity in wild type postdauer adults – is a bit skewed towards one point of view at the expense of an alternative (and arguably more valid) interpretation.The alternative view comes from considering the raw data in the dataset supporting Figure 1 (83372_0_data_set_1656224_q6n6fh.xlsx). In the dataset, it is apparent that the brood sizes of daf-12(lf) daf-9(lf) and daf-36(lf) non-dauer controls are much lower than wild type non-dauer controls. So, it appears that steroid hormone signaling is somehow required for full fecundity in non-dauer adults. Strikingly, the brood size defects of these mutants are essentially suppressed in postdauer animals (see 83372_0_data_set_1656224_q6n6fh.xlsx). Therefore, these results could be interpreted to mean that daf-12, daf-9, and daf-36 are actually more critical for full brood size during continuous development than they are during postdauer development. In other words, one could say that "… daf-12 signaling is required for full fecundity during continuous development, and is relatively dispensable during postdauer development." Similarly, Figure 4 shows that postdauer daf-12 mutants are suppressed for the reduced fat storage that these mutants display when developed continuously (without starvation or dauer arrest). Indeed, fat levels in postdauer daf-12 mutants appear to be restored to essentially the same levels as exhibited by postdauer wild type adults. This finding can be interpreted to suggest that daf-12 is not required for normal fat metabolism in postdauer animals (or at least daf-12 is less critical for fat storage in postdauer adults than in non-dauer adults). Note that this way of framing the conclusions from these data is the opposite to how the authors state their conclusions (that daf-12 regulates fat storage and fecundity of postdauer adults).

We appreciate your comments on the interpretation of our data. We had previously considered the possibility that passage through dauer rescues the brood size and intestinal fat storage of *daf-12*, *daf-9*, and *daf-36* mutants. Due to the fact that many strains we examined have reproductive defects in controls, we decided to only compare the CON and PD_Stv_ brood sizes within a strain. However, your comments have brought to our attention that our original interpretation was limited. If taken alone, your alternative interpretation that DA and/or DAF-12 is not required in postdauers, or that the phenotypes are rescued by passage through dauer seems valid. However, we think the *daf-12* data in its entirety suggests that DAF-12’s role in regulating these phenotypes is more nuanced. We argue that our data suggests that DAF-12 is contributing to the brood size and fat storage phenotypes based on the following observations.

– DAF-36 is required for the delayed germline proliferation phenotype in postdauer larvae, indicating that DA-dependent DAF-12 activity is required in early germline development.

*– Dhs-16* mutants do not have a significant reproduction defect, but also do not show the PD_Stv_ decrease in brood size, suggesting that DA-dependent DAF-12 activity is required. We have not examined early germline proliferation in this mutant to determine at what time in development it is required.

*– Daf-12* embryos have significantly increased fat compared to wildtype. At the least, this observation suggests that DAF-12 plays a role in regulating vitellogenesis.

– Finally, the *daf-12; tcer-1* and *daf-12; kri-1* double mutants exhibit or exacerbate the *daf-12* phenotype alone. This observation suggests that the DAF-12 activity, independent of DA, is contributing to the postdauer brood size phenotype. We have not yet examined if this is also the case for lipid storage.

Together, these data suggest that DAF-12 is playing a role, both DA-dependently and DA-independently, at different developmental timepoints to regulate PD_Stv_ phenotypes. We have modified the manuscript text to present both interpretations of our data in the Results section and included an argument for DAF-12 involvement in the Discussion section. We are further investigating the potential roles of DAF-12 in lipid metabolism and brood size regulation in *C. elegans* adults.

2) An instance where an additional experiment could have provided critical tests of otherwise relatively speculative interpretations.In this paper, following on from initial findings reported in Ow et al. 2018, the authors explore the phenomenon wherein postdauer animals can have altered numbers of germ cells at defined stages of larval development, suggesting differences in the timing of key steps in germline proliferation. The authors suggest that this altered germline proliferation program could affect brood size by altering the number of sperm available for self-fertilization. This is a very interesting hypothesis, that was unfortunately not tested directly by crossing the postdauer hermaphrodites to males to determine if sperm are indeed limiting.

We performed the experiment where CON and PD_Stv_ hermaphrodites were mated to control males and the number of progeny was counted. We found that the brood sizes of mated CON and PD_Stv_ hermaphrodites were statistically similar (see Appendix 1-figure 3).

3) Another instance where an additional experiment could have provided critical tests of otherwise relatively speculative interpretations.Another very interesting observation reported here is that the F1 progeny of postdauer adults, on average have more stored fat than F1 progeny of continuously-developed adults. This is the reverse of the P0 situation, where postdauer adults have less stored fat than controls (Figure 6). This inversion between P0s and F1s is absent in hrde-1 or prg-1 mutations, which are defective in RNAi inheritance. These data support the conclusion that small RNA mediated gene silencing could mediate the reprogramming of progeny physiology. What is missing is a demonstration of whether or not the reprogrammed F1 animals manifest any consequences from the reprogramming. In particular, Does the increased lipid storage in F1 animals confer upon them an altered fecundity and/or longevity?

We measured the longevity of CON and PD_Stv_ P_0_ and F_1_ populations. Interestingly, we found that PD_Stv_ adults have significantly increased longevity compared to CON adults, similar to what we observed for pheromone-induced postdauers (Hall et al. 2010). However, the increased longevity of postdauer adults was not inherited in the F_1_ progeny (see Figure 5C). Similarly, we tested whether the increased lipid storage in F_1_ progeny of PD_Stv_ adults altered their brood size, but the F_1_ of CON and PD_Stv_ adults had statistically similar brood sizes (Figure 5—figure supplement 3). We are continuing to examine whether the increased lipid storage in PD_Stv_ F_1_ progeny conveys a physiological advantage over CON progeny.

Reviewer #3:[…] Overall, the authors tackled an interesting and novel biological question using a system well established and pioneered by their lab. However, the data presented are somewhat disjointed, and at times seemingly contradict each other. Parts of the paper is difficult to follow and the conclusions are unclear.1. In general, many of the strongest effects on brood size from the genetic analysis (Figures 1-2) are occurring in strains with very small brood sizes to begin with. Thus, plotting the data as log2 fold change normalized between PDst adults and control risks visually exaggerating the effects and misleading the reader. Although the raw data are included and the reader can find out the extreme differences in brood sizes of the mutant strains (for example, daf-12 mutants), the lack of comments / discussion by the authors made the fold change figure misleading. The authors should consider plotting the brood size data (instead of just the fold change), so this difference can be easier to visualize. Although this does not necessarily negate the effects claimed by the authors, the authors need to clearly indicate that some of the strains have greatly reduced brood sizes when cultured under normal condition. Due to this caveat, the genetic interpretation is also unclear. One could argue that the transient dauer arrest somehow reverses the brood size defects of the mutants, rather than that the mutation suppresses the mild reduction on brood size caused by dauer arrest.

We would like to reassure the reviewer that our intention to present the data as log2 fold change between PD and CON adults was not to mislead the reader, but rather was an attempt to simplify the comparison of brood sizes within a strain. As suggested, we have remade all the figures using scatterplots representing brood size data to include both CON and PD_Stv_ data, and we used an ANOVA with *post hoc* test for multiple comparisons for statistical analysis. We have also included text describing the increase in brood size in *daf-12* PD_Stv_ adults. Please see the response to Reviewer #2, comment #1 for more details.

2. Similar to comment 1, the authors do not address why PDst should increase the brood size of the mutant strains. It would be logical that if a factor, like DAF-12, was required for the decrease in fertility of PDst adults that in this mutant strain there be no change in brood size after PDst, however in many cases the authors are actually seeing an increase in brood size rather than no change. Is this likely to be an artifact of the low brood size that these strains have under normal condition? It is interesting that the authors previously observed an increase in brood size in post-dauer adults forced into dauer by crowding, but this point is not discussed. Either way, the authors should discuss their interpretation of the brood size increase in PDst adult mutant strains in the text.

We have edited the text to include a comment regarding the similarity of increased postdauer brood size between pheromone-induced postdauer adults and some of the mutant strains examined. We have also added increased discussion regarding the *daf-*

*12* brood size phenotype. Please see the response to Reviewer #2, comment #1 for more details.

3. It would be interesting to investigate if there is any interaction between steroid signaling, TCER-1 and KRI-1 in regulating the low fecundity phenotype. That would help make the manuscript more coherent and easier to follow. In its current state, the earlier findings are not well-connected with the latter findings.

We had previously included the brood sizes of *daf-12; tcer-1* and *daf-12; kri-1* double mutants in Figure 1—figure supplement 1B, with a discussion of this experiment in the Appendix 1. We found that the brood sizes of these strains mimicked or exacerbated the *daf-12* alone brood size, suggesting that *daf-12* is acting in parallel or downstream of the reproductive longevity pathway. We argue that *daf-12* is acting downstream of *tcer-1* and *kri-1* (see Appendix 1). We have now included a statement of this experiment in the main manuscript text to bring it to the readers attention.

4. DAF-16 plays a critical role in regulating stress-mediated phenotypes. In fig-2B, the authors investigated the brood size of a strain that constitutively localizes DAF-16 in the nucleus. It would be helpful to include the brood size of daf-16(-) worms in PDstv/CONsv to definitely rule out the possibility of DAF-16 involvement.

*Daf-16* mutants are dauer defective. We attempted to force dauer formation using different alleles with no success.

5. At the beginning of the result section, in lines 111-117, the authors discuss the changes in gene expression that are shared between PDst adults and glp-1 germliness mutants, however they do not show the results of the comparison. Although the RNA-seq experiments were from a previous paper from the lab, the authors should still show the comparison results to convince readers that the similarities between PDst adults and glp-1 warrant investigation. For example, they could show a gene ontology analysis to highlight the pathways that are in common, or show Venn diagrams indicating significant overlap. Without the comparison data, it is difficult for the reader to assess how similar PDst adults and glp-1 pathways really are.

We did not perform a systematic analysis of PD_Stv_ RNA-Seq data with *glp-1* RNA-Seq data. However, *glp-1* animals have multiple characteristic gene expression signatures that have been shown to contribute to its longevity phenotype. We curated this list of gene expression changes from the literature for qualitative comparison with the gene expression changes observed in our PD_Stv_ RNA-Seq data. The list of genes with notable gene expression changes comparable to *glp-1* is found in Supplementary file 1. We have also performed a comparison of gene expression changes in PD_Stv_ with DAF-16, PQM-1, and TCER-1 targets, which can be found in Figure 2—figure supplement 1-Source Data 1 and Figure 2—figure supplement 1-Source Data 2.

6. The genetic data in Figure 3 was especially difficult to interpret, and the authors need to develop the discussion of this figure more thoroughly. Data shown in Figure 3A suggests that none of the factors (with the exception of sbp-1) show a robust suppression of the PDst fertility phenotype. This is also confusing because the authors include several different alleles of nhr-49 mutants, and see effects in some, but not all of the mutants. Moreover, some mutants are gain-of-function and some loss-of-function, but the authors have not explained why they would expect to see the same phenotype in both GOF and LOF alleles, so overall the results are difficult to interpret. The data become more convincing moving onto Figure 3C. Nevertheless, the authors need a more thorough discussion and consider the inconsistency in their data and interpret the results accordingly.

We have edited the discussion of Figure 3 to improve the clarity.

7. For Figure 3E, the authors state their hypothesis in the text (around line 274) that oleic acid (OA) supplementation should further decrease the brood size after PDst since they just found in Figure 3D that the loss of fat metabolism genes (that produce OA) could rescue the decreased brood size phenotype in PDst adults. This hypothesis is logical from their data, however the authors see the opposite of this result since all strains show an increase in brood size in Figure 3E when on OA. Of course surprising results occur, but the authors do not attempt to reconcile the discrepancy between their hypothesis and the results, leaving the reader very confused. They conclude that OA may be a limiting factor for PDst reproduction, but this is in contrast to their previous conclusion that reducing factors that produce OA with genetic mutants (hence limiting OA) would be beneficial to PDst reproduction. This discussion needs to be clarified. Additionally, the authors do not show the effect of OA on WT worms not exposed to PDst, which could be an important control and perhaps help the authors explain some of their contradictory findings.

We have edited the text to improve the clarity of this section. Please see Reviewer #1 comment #2 for more details.

8. Following comment 7, it might be helpful to determine whether induction of the fat genes indeed result in changes in MUFAs.

We were unable to perform this complicated experiment. However, we did perform fatty acid profiling for CON and PD_Stv_ adults (please see Figure 4—figure supplement 1C and Reviewer #1 concerns, comment #2). In addition, we tested whether the *fat* double mutants were required for the OA-dependent increase in PD_Stv_ brood size and found that *fat-7* is required for the phenotype. This result was surprising given that *fat-7* is required for OA biosynthesis, not OA metabolism. The manuscript text has been edited to discuss these results.

9. The authors show that PDstv adults have increased lipid storage in embryos, even in fat-6(-); fat-7(-) mutants. Do the authors consider increased lipid storage in embryos to be a general phenomenon of low lipid storage mutants?

We did not perform ORO staining in *fat* mutants. However, we did perform ORO staining in *daf-12* mutants. The levels of fat storage in *daf-12* CON and PD_Stv_ embryos were similar, despite the observation that *daf-12* CON adults have significantly decreased fat storage compared to *daf-12* PD_Stv_. Based on these results, we hypothesize that the levels of vitellogenesis are likely independent of the intestinal fat levels.

10. It may be interesting to check the expression of the desaturases in daf-12(-), tcer-1(-) and kri-1(-) worms to tie in the role of these desaturases in regulating low fecundity phenotype. This will also help to connect the earlier results with the fat phenotype.

Unfortunately, we were unable to include these results in this manuscript and we felt it went beyond the scope of this study. However, this question is actively being investigated for our next manuscript.

11. The authors observed that depletion of VIT-1 resulted in significantly increased intestinal fat deposit (Figure 4C and 4D). Since vitellogenins are known to carry fat from the intestine to the developing oocytes, its depletion would be expected to stop the transfer of fat from the intestine to the oocytes, which could result in increased intestinal fat. The authors should consider including this interpretation in the text.

We have edited our manuscript to include this interpretation of our results.

12. The authors suggested that germ cell numbers in daf-12(-) and tcer-1(-) could be connected to their brood sizes in postdauer worms. However, since the difference in germ cell numbers appears to be very small, and the change in brood size appears to be very large. Authors should comment on that.

We agree with the reviewer that the changes in germ cell rows in larvae do not correlate with adult brood sizes in *daf-36* and *tcer-1* mutants. These mutant strains have characterized defects in gonad development that are likely contributing to the decreased brood sizes observed in adulthood. We hypothesize that DAF-12 contributes to multiple aspects of PD_Stv_ reproduction by acting in different tissues at distinct developmental time points. We have rewritten part of our Discussion to include this argument.

13. Lastly, the paper at its current state does not read like a coherent paper, but rather it reads like two separate sections.

To improve the coherence of the manuscript, we have reversed the order of Figures 6 and 7 in order to keep the results regarding lipid content together. We have also extensively edited the text in order to improve the clarity.